# State-level macro-economic factors moderate the association of low income with brain structure and mental health in U.S. children

David G. Weissman [1] ✉, Mark L. Hatzenbuehler[1], Mina Cikara [1],
Deanna M. Barch [2] & Katie A. McLaughlin[1]

Macrostructural characteristics, such as cost of living and state-level anti-poverty programs relate to the magnitude of socioeconomic disparities in brain development and mental health. In this study we leveraged data from the Adolescent Brain and Cognitive Development (ABCD) study from 10,633 9-11 year old youth (5115 female) across 17 states. Lower income was associated with smaller hippocampal volume and higher internalizing psychopathology. These associations were stronger in states with higher cost of living. However, in high cost of living states that provide more generous cash benefits for low-income families, socioeconomic disparities in hippocampal volume were reduced by 34%, such that the association of family income with hippocampal volume resembled that in the lowest cost of living states. We observed similar patterns for internalizing psychopathology. State-level anti-poverty programs and cost of living may be confounded with other factors related to neurodevelopment and mental health. However, the patterns were robust to controls for numerous state-level social, economic, and political characteristics. These findings suggest that state-level macrostructural characteristics, including the generosity of anti-poverty policies, are potentially relevant for addressing the relationship of low income with brain development and mental health.

Adults who were raised in families with lower income as children have lower educational attainment, are more likely to rely on public assistance, and tend to have more mental and physical health problems than those raised in higher income families[1–3]. Increasing evidence demonstrates that family income is associated with structural differences in the developing brain[4–8], which may contribute to these disparities in later-life outcomes. Numerous environmental processes may contribute to these patterns. Financial hardship constrains the time, material, cognitive, and emotional resources caregivers have to dedicate to their children and is associated with higher exposure to stressful life events (e.g., violence), chronic

stressors (e.g., food insecurity), and a less predictable environment[9,10]. Macrostructural characteristics that influence the material resources available to low-income families may relate to the strength of the association between low income and health and neurodevelopmental outcomes. This study usesdata from the national multisite Adolescent Brain and Cognitive Development (ABCD) Study of children in the United States to evaluate whether macrostructural characteristics of U.S. states, including the generosity of anti-poverty policies and cost of living, moderate the associations of low income with brain structure and mental health in early adolescence.

[1]Department of Psychology, Harvard University, Cambridge, MA, USA. [2]Department of Psychological & Brain Sciences, Washington University, St. Louis, MO, USA. ✉e-mail: dweissman@fas.harvard.edu

Family income is consistently associated with children's brain structure[4-8]. In particular, smaller hippocampal volume among children from families with lower income is a well replicated finding[7,11-15]. One likely explanation for this link is that increased exposure to stressors among children growing up in low-income contexts contributes to reduced hippocampal volume[2,9]. Indeed, low family income is associated with exposure to more stressful life events—such as neighborhood violence, conflict, and caregiver separation—in childhood[9,16]. In animal models, chronic stress has toxic effects on hippocampal neurons, leading to reductions in dendritic branching and neuronal loss[17-19]. Consistent with this hypothesis, greater exposure to stress mediates the association of family income with children's hippocampal volume[15]. Similarly, family income is consistently associated with higher levels of internalizing and externalizing psychopathology in children and adolescents[2,3,20].

However, broad social and economic factors relate to the strength of the associations between low income, mental health, and neural outcomes. For example, in the U.S. the federal poverty line is used to determine which families are eligible for federal resources aimed at supporting families with low income. While this metric assumes that the costs of meeting basic needs are the same across the U.S., the cost of living in fact varies widely, leading to geographical disparities in the value of a dollar. For example, an item that costs $1.10 in California costs 92 cents in Oklahoma based on regional price parity[21]. Thus, living in a region with a high cost of living may enhance financial strain for families with low income and magnify the impact of low income on children's hippocampal volume and mental health.

Conversely, the generosity of the social safety net for low-income families, reflected in public policies designed to help families in poverty, may lessen the impact of low income on children's hippocampal volume and mental health. Government sponsored anti-poverty programs are intended to ensure that families have the ability to provide for basic necessities such as food, shelter, and medical care. These programs include cash assistance programs like the Earned Income Tax Credit (EITC) and Temporary Assistance for Needy Families (TANF), as well as programs that provide benefits that meet specific needs, like the health insurance provided to low-income families by Medicaid. The EITC provides a tax credit to increase the income of working families below a certain income threshold, and TANF provides temporary cash assistance to families that are out of work. More generous and accessible cash benefits are associated with better family functioning, physical health, academic achievement, and overall wellbeing for children in families who receive benefits relative to those who do not (assessed via comparisons generated by geographic or temporal differences in policy adoption)[22-24]. As another example, the Affordable Care Act funded expanded access to health insurance through Medicaid to all U.S. citizens with income up to 138% of the federal poverty line, although not all states adopted these expanded benefits. Parents in states where Medicaid was expanded report reduced problems paying medical bills and lower psychological distress than those in states that did not expand Medicaid[25]. However, the availability and generosity of these three anti-poverty programs varies widely across U.S. states. Importantly, other public policies may also mitigate the impact of low income on child development[26], but their generosity does not vary between U.S. states. These policies include the Supplemental Nutrition Assistance Program (i.e., food stamps) and, recently, the child tax credit. In addition, policies at the county, city, or school district level may also contribute to an environment that lessens socioeconomic disparities.

Until recently, it has been challenging to examine whether macrostructural characteristics, such as anti-poverty policies, moderate the association between family income and neural outcomes. The ability to examine this question has been hindered by the fact that most neuroimaging studies are conducted in a single community, such that all participants within each study are exposed to the same macro-social context (precluding comparisons across contexts). However, two recent studies suggest that macrostructural characteristics, such as anti-poverty policies, may be associated with neurodevelopmental outcomes. For example, the associations of income with total brain volume and cognitive outcomes were weaker in studies conducted in Europe than in the U.S[27]. This finding could be related to macro-social and policy differences across these contexts; however, the researchers did not directly test that explanation. More relevant, 1-year-olds whose families received unconditional cash payments of $333 a month in a large randomized controlled trial exhibited greater power in high-frequency bands during electroencephalography, a pattern that is associated with later cognitive skill development[28]. This finding suggests that increased income of a magnitude similar to that provided by anti-poverty cash assistance programs may have a positive influence on neurodevelopment.

The current study builds on this recent work to examine whether cost of living and the generosity of a state's social safety net for low-income families moderate the association of family income with hippocampal volume and mental health outcomes using data from the large, multisite ABCD study. This study is well suited to answering our research question because it sampled respondents from across 21 sites (in 17 states) that differed in terms of their cost of living and anti-poverty policy climates. Using these data, we examine the associations of family income with hippocampal volume and mental health and determine whether state-level macrostructural characteristics—including cost of living and the generosity of the three largest anti-poverty programs that vary by state (i.e., EITC, TANF, and Medicaid)—moderate the association between family income and those outcomes. These programs have documented efficacy at improving family functioning, physical health, academic achievement, and overall wellbeing and reducing psychological distress based on natural experiments[22-25]. More generous anti-poverty policies may also help to mitigate the impacts of low income on children's neural development and mental health in states where those policies are implemented, leading to reduced socioeconomic disparities. Importantly, the correlational analyses described in this study are not intended to serve as a direct evaluation of these specific policies. Rather, the policies selected in this study are intended to serve as quantitative indicators of the broader macrostructural environment related to the generosity of the state's social safety net for families living in poverty, which we are able to operationalize on a meaningful scale (e.g., dollars) and at a meaningful level of regional variability (U.S. states) at which these policies are actually implemented.

We hypothesized that lower family income would be associated with smaller hippocampal volume and higher internalizing and externalizing problems, consistent with prior work[2,3,7,11-15,20,29]. We additionally hypothesize that the magnitude of those associations would be greater in states with higher cost of living, but would be smaller in states with more generous anti-poverty programs, particularly in high cost of living states. In doing so we provide a test of whether the well-established associations of family income with brain structure and mental health vary as a function of the broader macrostructural environment in which children are being raised. Importantly, for any significant interactions among family income, anti-poverty programs, and cost of living, we also evaluated at what levels of income cost of living and anti-poverty programs were associated with hippocampal volume and psychopathology symptoms. The presence of moderation among participants who were eligible for anti-poverty benefits, combined with a lack of moderation among non-eligible respondents, would help bolster inferences that it may be the generosity of anti-poverty policies that are reducing the association of family income with hippocampal volume and mental health. Finally, in order to rule out plausible alternative explanations and to reduce spurious contextual influences, we examined whether our results were robust to controls for a wide range of state-level social, economic, and political

characteristics that may be associated with the generosity of state anti-poverty programs and thus may operate as confounders. These characteristics included population density, economic conditions of the state (economic inequality, unemployment rate), non-economic characteristics reflecting the social and political climate in the state (political preferences, women's political participation, reproductive rights, incarceration rate, tightness/looseness—i.e., cultural differences around rule and norm adherence), and characteristics reflecting equity in the education system (state-funded preschool enrollment, reading proficiency among students from low-income backgrounds).

In this work, we show that state-level macrostructural characteristics, specifically cost of living and the generosity of anti-poverty policies, moderate the association of low family income with hippocampal volume and internalizing problems among 9–11-year-old youth in the US. This suggests that these macrostructural factors are relevant for addressing the relationship of low income with neurodevelopmental and mental health outcomes.

## Results

### Family income, hippocampal volume, and psychopathology

Higher family income—as reflected in log-income-to-needs ratio—was positively associated with hippocampal volume when controlling for total intracranial volume, age, and sex ($t(9,888) = 7.78$, $p < 0.001$, $B = 62.71$, 95% CI = 46.91 to 78.52, $\beta = 0.079$), such that hippocampal volume was larger for participants with higher family income. However, there was considerable variability in both the intercept of hippocampal volume (*SD of random intercepts* = 69.92) and in the slope of the association between family income and hippocampal volume (*SD of random slopes* = 21.34) across the 21 ABCD sites (Fig. 1). Model residuals were normally distributed, homoscedastic, and independent.

Family income was negatively associated with internalizing ($t(10,605) = -3.97$, $p < 0.001$, $B = -0.59$, 95% CI = −0.88 to −0.030, $\beta = -0.058$) and externalizing ($t(10,605) = -6.46$, $p < 0.001$, $B = -1.50$, 95% CI = −1.96 to −1.05, $\beta = -0.152$) problems, again with considerable variability in the intercepts and slopes of internalizing (SD = 1.41 and 0.477, respectively) and externalizing (SD = 1.57 and 0.950, respectively) problems across the 21 ABCD sites. Model residuals were normally distributed, homoscedastic, and independent.

### State-level moderators of family income-hippocampal volume associations

Cost of living and the generosity of state anti-poverty programs—including two cash assistance programs (EITC and TANF) and the presence of Medicaid expansion—were investigated as potential moderators of the association of family income with youth hippocampal volume. Cash assistance was operationalized as the mean of the average monthly EITC and TANF benefit in each state. Medicaid expansion was a dichotomous variable indicating whether or not the state had expanded Medicaid benefits. State-level characteristics are summarized in Table 1.

Results of moderation analyses are summarized in Table 2. We observed a 3-way interaction between family income, cost of living, and generosity of cash assistance programs in predicting hippocampal volume ($t(9,883) = -3.14$, $p = 0.002$, $B = -3.64$, 95% CI = −5.91 to −1.36). Model residuals were normally distributed, homoscedastic, and independent. Decomposing this interaction revealed that low-income participants living in states with high cost of living (i.e., states with a cost of living 1 SD above the mean, equivalent to a regional price parity of 1.07) and high cash benefits (i.e., states with cash assistance 1 SD above the mean, equivalent to a mean monthly cash benefit of $531) have hippocampal volumes that are on average 60 mm³ larger than low-income participants living in states with high cost of living (i.e., states with a cost of living at the mean, equivalent to a regional price parity of 1.01) and low cash benefits (i.e., states with cash assistance 1 SD below the mean, equivalent to a mean monthly cash benefit of

$317) (Fig. 2). Of these states with low cash benefits, none had cost of living that was 1 SD above the mean, so mean cost of living was used for this comparison to avoid extrapolating beyond the true range of the data, making these estimates conservative. On average, the difference in hippocampal volume between low- and high-income participants in high cost of living states with low cash benefits is 195 mm³, but only 129 mm³ in states where cost of living and cash benefits are both high. Thus, more generous cash benefits at the state level are associated with income disparities in hippocampal volume that are about 34% lower in states with high (vs. low) cost of living. Importantly, a simple slopes analysis demonstrates that the interaction between cost of living and cash benefits is only significant when log-income-to-needs ratio is low −about 1 SD below the mean, equivalent to about 80% of the poverty line. This finding suggests that cost of living and the generosity of anti-poverty policies are associated with hippocampal volume only for children in low-income families who are eligible for the benefits of these programs.

There was a similar 3-way interaction between family income, cost of living, and Medicaid expansion in relation to hippocampal volume ($t(9883) = -3.31$, $p = 0.001$, $B = -983$, 95% CI = −1565 to −400). Decomposing this interaction revealed that among children living in states with higher cost of living, the association between family income and hippocampal volume was weaker in states that expanded Medicaid (Fig. 3) than in states that did not. In high cost of living states (1 SD above the mean), the disparities in hippocampal volume between high (1 SD above the mean) and low (1 SD below the mean) income participants were 43% smaller in states that expanded Medicaid than in states that did not.

There were no significant 2-way interactions between family income and cost of living, nor between family income and Medicaid expansion in relation to hippocampal volume. The 2-way interaction between cash assistance and family income was significant when the model also controlled for the interaction between family income and cost of living (Supplemental Table S1). In a supplemental analysis, we examined the 3-way interactions of family income, cost of living, and the state minimum wage—an alternative indicator of the relative generosity of the state's social safety net for low-income families—in relation to hippocampal volume and found the interaction was significant and the pattern of results was the same as for cash assistance and Medicaid expansion (Supplemental Table S2).

### State-level moderators of family income-psychopathology associations

We also examined whether cost of living and the generosity of anti-poverty programs moderated the association between family income and adolescent internalizing and externalizing problems. We observed 3-way interactions among family income, cost of living, and both the generosity of cash assistance programs ($t(10,602) = 2.67$, $p = 0.008$, $B = 0.0615$, 95% CI = 0.0163 to 0.107) and Medicaid expansion ($t(10,602) = 2.43$, $p = 0.015$, $B = 14.9$, 95% CI = 2.90 to 27.0) in predicting internalizing problems. Model residuals were normally distributed, homoscedastic, and independent. Decomposing this interaction revealed that among children living in states with a higher cost of living, the association between family income and internalizing problems was lower in states with more generous anti-poverty cash assistance programs (Fig. 4a) and in states that expanded Medicaid (Fig. 4b) relative to states with less generous cash assistance and that did not expand Medicaid, respectively. On average, the difference in internalizing problems t-scores between low- and high-income participants in high cost of living states with low cash benefits is 2.11, but only 1.08 in states where cost of living and cash benefits are both high. Thus, more generous cash benefits are associated with income disparities in internalizing symptoms that are approximately 48% lower in high (vs. low) cost of living states. A simple slopes analysis demonstrates that the interaction between

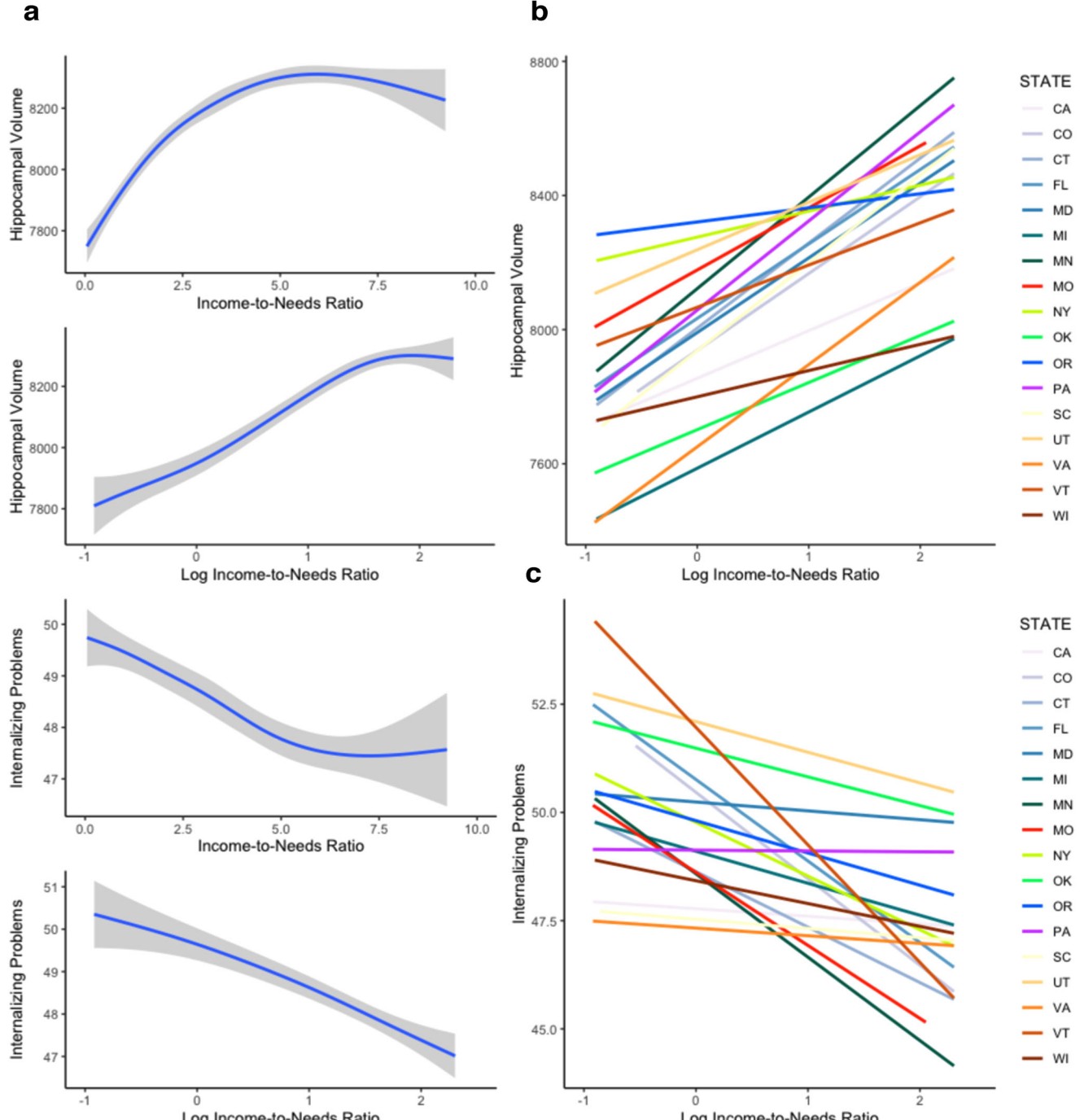

**Fig. 1 | The associations of family income with hippocampal volume and internalizing problems. a** Plots illustrate that log-transformed income-to-needs ratio better characterizes the association between family income and both hippocampal volume and internalizing problems in the ABCD sample than raw income-to-needs ratio. Error bands reflect the 95% confidence intervals of these estimates.

**b** Variability between states in the strength of the linear association between log-income-to-needs ratio and hippocampal volume ($N = 9913$). **c** Variability between states in the strength of the linear association between log-income-to-needs ratio and internalizing problems ($N = 10,633$, bottom).

cost of living and cash benefits is not significant at any level of log-income-to-needs ratio present in the data. However, when income is low, the cash benefit by cost of living interaction is negative, such that in higher cost of living states, more generous cash benefits are associated with greater reductions in internalizing problems. Conversely, when income is high, the cash benefit x cost of living interaction is positive, suggesting that cost of living and the generosity of anti-poverty policies are associated with lower internalizing problems only for children in low-income families who are eligible for these benefits relative to children in high-income families who are not.

There was no similar 3-way interaction between family income, cost of living, and the generosity of anti-poverty programs in relation to externalizing problems (Table 2). There were no significant 2-way interactions between family income and cost of living or between family income and the generosity of anti-poverty programs in relation to internalizing or externalizing psychopathology symptoms (Supplemental Table S1). As with hippocampal volume, we conducted

**Table 1 | Summary of state-level variables and sample characteristics**

| State | N | % Black | % Latinx | % White | RPP | EITC | TANF | Mean cash | Expanded medicaid |
|---|---|---|---|---|---|---|---|---|---|
| California | 1934 | 9.5 | 50.5 | 62.7 | 1.10 | 0.45 | 785 | 585 | Yes |
| Colorado | 565 | 8.5 | 22.3 | 92.2 | 1.02 | 0.10 | 508 | 400 | Yes |
| Connecticut | 634 | 27.6 | 22.9 | 68.9 | 1.06 | 0.23 | 698 | 513 | Yes |
| Florida | 1085 | 24.3 | 46.3 | 73.2 | 1.01 | 0 | 303 | 285 | No |
| Maryland | 605 | 46.6 | 7.2 | 52.2 | 1.07 | 0.50 | 709 | 554 | Yes |
| Michigan | 722 | 27.7 | 9.0 | 73.8 | 0.93 | 0.06 | 492 | 387 | Yes |
| Minnesota | 607 | 8.7 | 5.7 | 91.1 | 0.97 | 0.35 | 532 | 446 | Yes |
| Missouri | 705 | 36.6 | 3.6 | 66.4 | 0.92 | 0 | 292 | 279 | No |
| New York | 339 | 29.5 | 14.3 | 71.7 | 1.10 | 0.30 | 789 | 567 | Yes |
| Oklahoma | 743 | 24.0 | 19.6 | 76.3 | 0.92 | 0.05 | 292 | 286 | No |
| Oregon | 585 | 8.0 | 16.3 | 88.0 | 1.01 | 0.09 | 506 | 398 | Yes |
| Pennsylvania | 455 | 63.7 | 3.6 | 41.8 | 0.99 | 0 | 421 | 344 | Yes |
| South Carolina | 379 | 34.6 | 3.8 | 66.2 | 0.94 | 0.42 | 286 | 332 | No |
| Utah | 1002 | 2.9 | 10.8 | 94.8 | 0.99 | 0 | 498 | 382 | No |
| Virginia | 552 | 34.6 | 7.9 | 67.6 | 1.01 | 0.20 | 442 | 381 | Yes |
| Vermont | 578 | 3.1 | 3.0 | 94.6 | 1.01 | 0.36 | 640 | 501 | Yes |
| Wisconsin | 385 | 17.7 | 9.9 | 86.5 | 0.93 | 0.11 | 653 | 474 | No |

*RPP* regional price parity, *EITC* state-level earned income tax credit as a proportion of the federal EITC, *TANF* monthly temporary assistance for needy families' benefit in dollars, % Black, % Latinx, and % White represent the racial and ethnic makeup of the ABCD sample within that state.

supplemental analyses examining the 3-way interactions of family income, cost of living, and the state minimum wage in relation to internalizing and externalizing problems, and found the interaction was significant in both analyses (see Supplemental Table S2).

**Sensitivity analyses**

To determine if heterogeneity in the state-level associations between family income and study outcomes (i.e., hippocampal volume and internalizing problems) was specifically related to cost of living and the generosity of anti-poverty programs, and not to other potentially correlated state-level characteristics that do not as plausibly or specifically benefit low-income families, we conducted sensitivity analyses. Each analysis controlled for one of 10 state-level social, economic, and political characteristics that may serve as alternative explanations for these patterns. These characteristics included population density, economic inequality, unemployment rate, tightness/looseness, political preferences, women's political participation, reproductive rights, incarceration rate, state-funded preschool enrollment, and reading proficiency among students from low-income backgrounds. A detailed description of each measure is provided in Supplemental Materials. Our reported findings were robust to controlling for each of these wide-ranging state-level characteristics, as well as their interactions with family income (see Supplemental Tables S3 and S4 for details).

To further interrogate whether the patterns of findings were observed only for participants with lower income who could potentially benefit from these programs but not those with higher incomes who could not. we conducted a supplementary analysis, where we examined income dichotomously to reflect whether family income was above vs. below the federal poverty line. This analysis allowed us to examine Indeed, the pattern of results was consistent with our original findings, in that the interactions of cost of living, generosity of anti-poverty policies, and family income are all significant. However, if we use a higher cutoff that does not correspond with eligibility for benefits (i.e., dichotomizing above vs. below 5x the federal poverty line), these interactions are not significant for any outcome (Supplemental Table S5), providing further support for our interpretation that these associations are only apparent for individuals who are actually eligible for anti-poverty programs.

In addition, to evaluate whether results were driven by states with multiple sites and a greater number of participants (i.e., California and Florida), we reran our analyses including only the site with the most participants in each state (and therefore excluding 1 site in Florida and 3 sites in California). The results of these analyses were consistent with our original findings (See Supplemental Table S6).

## Discussion

We found that lower family income is associated with smaller hippocampal volume and greater internalizing and externalizing psychopathology in early adolescence, replicating prior findings in smaller samples[7,11–15,20] and previous analyses of a smaller portion of the ABCD sample[30]. Critically, however, we demonstrate that the magnitude of these associations varies as a function of state-level macrostructural characteristics—including the cost of living and generosity of anti-poverty programs. The disparities between high- and low-income participants in hippocampal volume and internalizing problems were 34% to 48% smaller, respectively, in states that had a high cost of living but that provided more generous benefits for lower-income families, as compared to high cost of living states with less generous benefits. In high cost of living states where anti-poverty programs were more generous, the association of family income with hippocampal volume and internalizing problems resembled that of the lowest cost of living states. Importantly, these associations were robust to controls for other state-level social, economic, political, and educational factors—including population density, unemployment rate, political preferences, and state-funded preschool enrollment—thereby ruling out some alternative explanations. These controls capture a diverse set of macrostructural characteristics that covary geographically with cost of living and the generosity of anti-poverty programs, with varying degrees of relevance for children and families with low income. Together, these findings suggest that macrostructural factors related to the generosity of the state's social safety net for families living in poverty are associated with socioeconomic disparities in hippocampal volume and internalizing problems, and that structural policy interventions may be an effective strategy for reducing these disparities, though this interpretation awaits replication with other research designs (e.g., quasi-experiments).

**Table 2 | Results of moderation analyses**

| Cash benefits | | | | Medicaid expansion | | | |
|---|---|---|---|---|---|---|---|
| **Hippocampal volume** | | | | | | | |
| | B | SE | p | | B | SE | p |
| Income | 82.0 | 9.66 | <0.001 | Income | 95.8 | 16.0 | <0.001 |
| Cost of living (COL) | −134 | 435 | 0.761 | COL | −278 | 614 | 0.657 |
| Cash benefit (Cash) | −0.032 | 0.217 | 0.885 | Medicaid expansion (ME) | −15.4 | 40.8 | 0.712 |
| Income x COL | 242 | 174 | 0.164 | Income x COL | 736 | 259 | 0.005 |
| Income x cash | −0.0992 | 0.101 | 0.326 | Income x ME | −26.8 | 18.6 | 0.149 |
| COL x cash | 2.63 | 2.84 | 0.370 | COL x ME | 350 | 649 | 0.599 |
| Income x COL x cash | −3.64 | 1.16 | 0.002 | Income x COL x ME | −983 | 297 | 0.001 |
| **Internalizing problems** | | | | | | | |
| Income | −0.936 | 0.187 | <0.001 | Income | −1.05 | 0.338 | 0.002 |
| COL | 2.92 | 10.18 | 0.778 | COL | 12.10 | 15.6 | 0.451 |
| Cash | −.00404 | 0.00572 | 0.492 | ME | −1.41 | 1.12 | 0.231 |
| Income x COL | .972 | 3.38 | 0.774 | Income x COL | −8.28 | 5.40 | 0.125 |
| Income x Cash | −.00027 | 0.00196 | 0.892 | Income x ME | 0.307 | 0.387 | 0.428 |
| COL x Cash | .0161 | 0.0803 | 0.844 | COL x ME | −11.6 | 18.0 | 0.532 |
| Income x COL x Cash | .0615 | 0.0231 | 0.008 | Income x COL x ME | 14.9 | 6.14 | 0.015 |
| **Externalizing problems** | | | | | | | |
| Income | −1.78 | 0.316 | <0.001 | Income | −1.65 | 0.580 | <0.001 |
| COL | −0.033 | 7.85 | 0.997 | COL | 8.80 | 11.7 | 0.464 |
| Cash | −0.00321 | 0.00441 | 0.479 | ME | −1.11 | 0.831 | 0.202 |
| Income x COL | 5.41 | 5.84 | 0.354 | Income x COL | −3.18 | 9.40 | 0.735 |
| Income x Cash | −0.00135 | 0.00336 | 0.688 | Income x ME | −.213 | 0.660 | 0.747 |
| COL x Cash | −0.0161 | 0.0618 | 0.799 | COL x ME | −12.5 | 13.4 | 0.370 |
| Income x COL x Cash | 0.0477 | 0.0409 | 0.243 | Income x COL x ME | 14.8 | 10.6 | 0.162 |

Analyses were conducted using linear mixed-effects models with the nlme package in R version 4.0.0 using two-tailed tests. Age, sex, and the proportion of participants at each site that were White, Black, and Latinx were also included as covariates in all analyses. Total intracranial volume and scanner type were included as covariates in models with hippocampal volume as the outcome. Reproducible code and full model output for all analyses is included in supplemental materials.

One reason why cost of living and the generosity of anti-poverty programs may be associated with differences in the strength of the associations among family income, hippocampal volume, and mental health is because they may amplify or reduce stressors associated with low income. Having greater financial resources may shield families from experiencing some of the chronic stressors associated with low income that can influence hippocampal development[15,16,31]. Further, by increasing financial resources and access to healthcare (i.e., in states that expanded Medicaid benefits), it is plausible that more generous anti-poverty programs could decrease the negative impact of some stressful life events on hippocampal volume and internalizing problems. Cumulative stress exposure is by no means the only potential process linking family income with neurodevelopment and mental health that might be impacted by the generosity of anti-poverty programs, however. For example, low income is also associated with lower cognitive stimulation, less supervision by adults, and a less predictable environment[9,32,33]. An ongoing experimental study that randomized families to receive cash benefits[34] has already found that those benefits may contribute to alterations in neural function[28] and thus may shed light on the role of these processes in explaining why the generosity of anti-poverty programs is associated with differences in the strength of the associations between low family income, neurodevelopment, and mental health.

Given that we observed the same type of 3-way interaction of family income and cost of living with cash assistance and Medicaid expansion, we expect that these policy differences—while helpful in and of themselves—are likely indicative of a broader collection of policies and structural supports for families facing economic hardship that were not directly measured in this study but that reflect the generosity of the state's social safety net for low-income families (e.g., availability of free or reduced cost early childhood education programs). Indeed, in supplemental analyses, we examined the 3-way interactions of family income, cost of living, and the state minimum wage—another indicator of the generosity of the state's social safety net for families with low income—in relation to hippocampal volume, internalizing problems, and externalizing problems, and found a similar pattern of results. In addition, the study design only allows us to evaluate whether anti-poverty programs that differ between states may mitigate socioeconomic disparities in hippocampal volume and mental health. Programs such as SNAP, which are administered uniformly across states are also associated with reductions in adverse childhood experiences and improved health and educational outcomes[26,35,36].

These findings have broad methodological implications for research on environmental factors in psychology and cognitive neuroscience. Most psychology and neuroimaging studies are conducted in a single community. In such designs, all participants experience the same macro-social context, which requires a focus solely on individual- and interpersonal-level explanations for observed associations. However, our results, together with other recent studies[37], demonstrate that the magnitude of individual-level associations between environmental factors, like family income, and developmental outcomes, like brain structure and psychopathology, vary systematically as a function of characteristics of the macro-social context. This context therefore demands consideration when comparing results collected in different communities and may provide an explanation for replication failures in studies focused on associations measured solely at the individual level. The degree to which different aspects of macro-social context

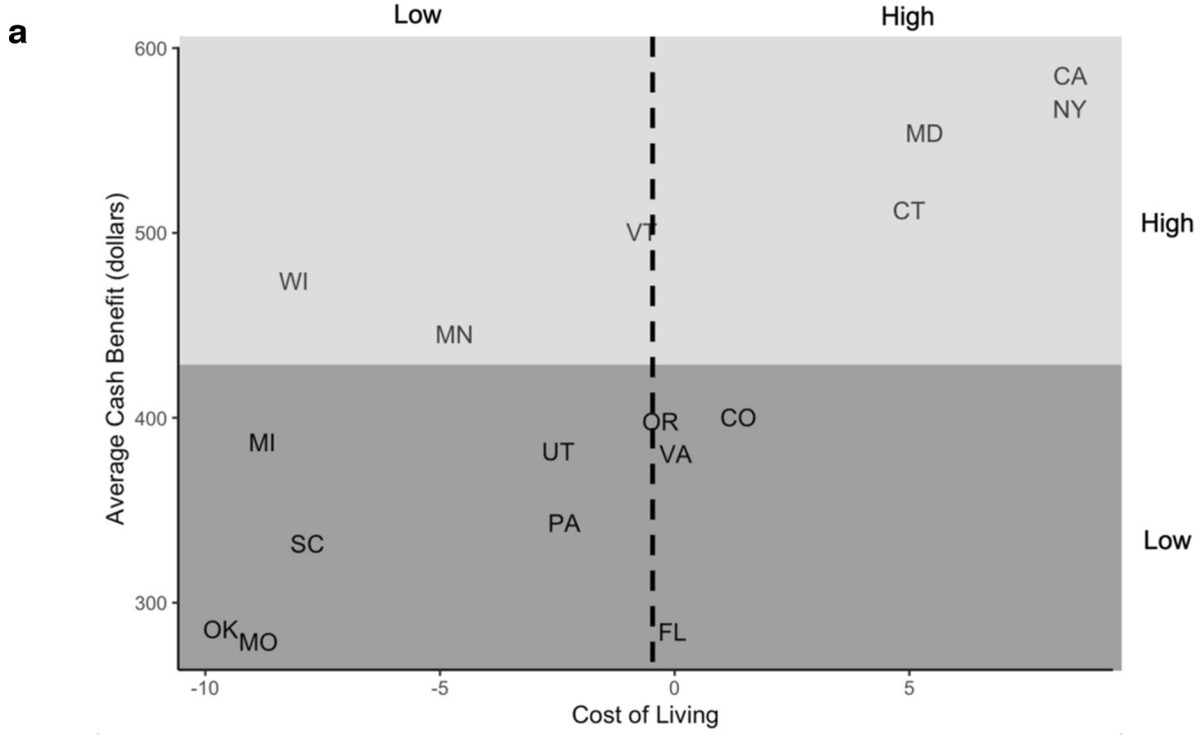

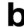

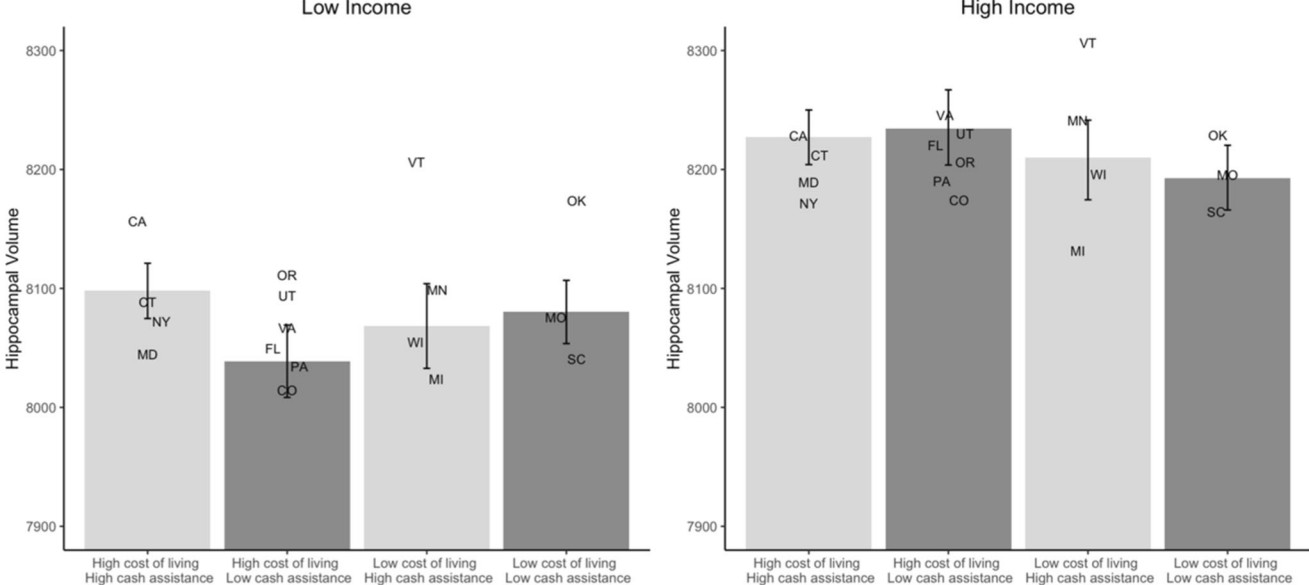

**Fig. 2 | The 3-way interactions among log-income-to-needs ratio, cost of living, and cash assistance in relation to hippocampal volume (*N* = 9913). a** The distribution of cost of living and generosity of anti-poverty programs were correlated across the 17 states. **b** The simple slopes are plotted to reflect the values actually present in the distribution of cost of living and cash benefits in the ABCD data. The hippocampal volume estimate for low cost of living, low cash benefit states is based on the intercept estimates for hippocampal volume when log-income-to-needs ratio is 1 SD above or below the mean, and cost of living and cash benefits are both 1 SD below the mean. The estimate for high cost of living, high cash benefit states is based on the intercept estimates when cost of living and cash benefits are both 1 SD above the mean. The estimate for high cost of living, low cash benefit states is based on the intercept estimates when cost of cost of living is at the mean, and cash benefits are 1 SD below the mean, as no state with cost of living 1 SD above the mean had low cash benefits in our data. The estimate for low cost of living, high cash benefit states is based on the intercept estimates when cost of living is 1 SD below the mean, and cash benefits are at the mean, as no state with low cost of living had cash benefits that were 1 SD above the mean in our data. This is illustrated in supplemental Fig. S1. Error bars reflect the 95% confidence intervals of these estimates. The random intercepts for each state when log-income-to-needs ratio is 1 SD above or below the mean are represented by that state's abbreviation.

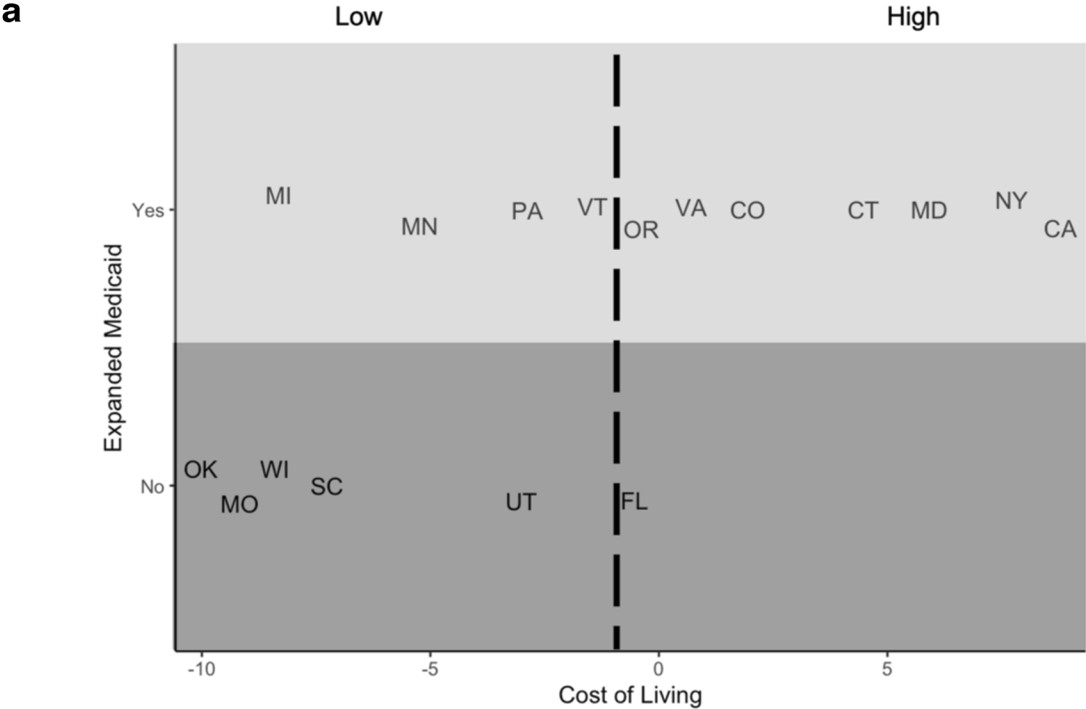

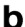

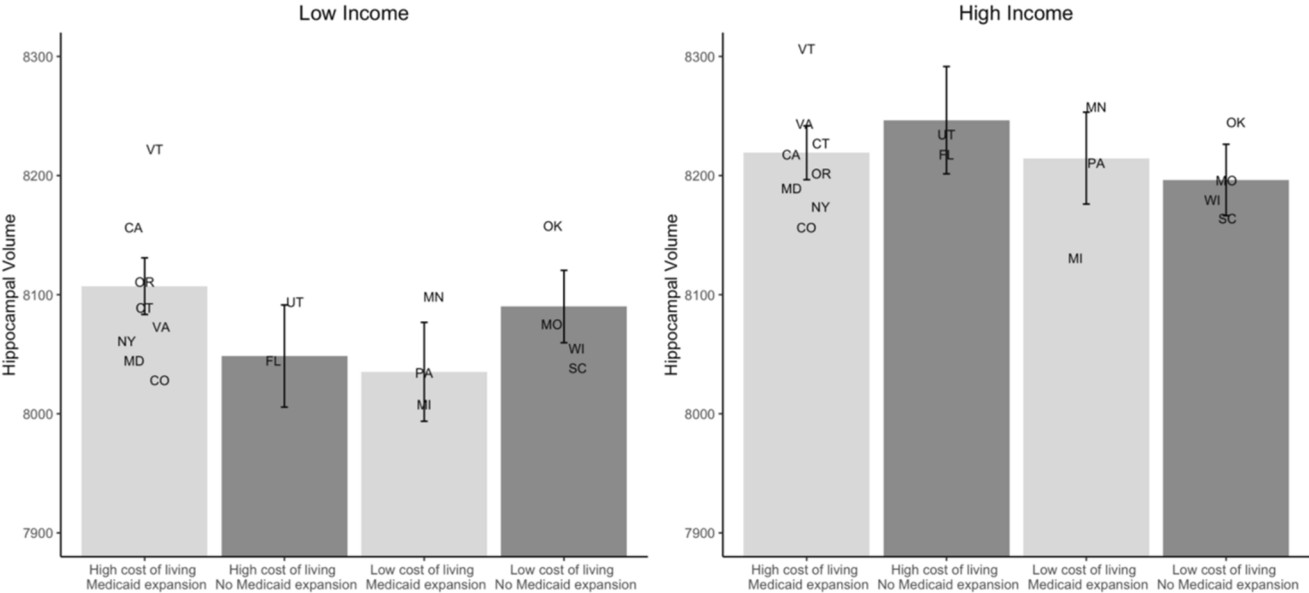

**Fig. 3 | The 3-way interactions among log-income-to-needs ratio, cost of living, and Medicaid expansion in relation to hippocampal volume (*N* = 9913). a** The distribution of cost of living and Medicaid expansion in the 17 states. The simple slopes are plotted to reflect the values actually present in the distribution of cost of living and cash benefits in the ABCD data. **b** The hippocampal volume estimate for states that did not expand Medicaid is based on the intercept estimates for hippocampal volume when log-income-to-needs ratio is 1 SD above or below the mean, and cost of living is 1 SD below the mean. The estimates for high cost of living states that expanded Medicaid are based on the intercept estimates when cost of living is 1 SD above the mean. The estimate for high cost of living states that did not expand Medicaid is based on the intercept estimates when cost of cost of living is at the mean, as no states with income 1 SD above the mean did not expand Medicaid. This is illustrated in supplemental Fig. S2. Error bars reflect the 95% confidence intervals of these estimates. The random intercepts for each state when log-income-to-needs ratio is 1 SD above or below the mean are represented by that state's abbreviation.

moderate other individual-level associations is an important question that could be explored in many additional ways in the ABCD data.

Some limitations are worth noting. First, this is a cross-sectional and observational study. However, it is implausible that children's hippocampal volume could alter their family's income or state-level characteristics, and simple slopes analyses suggested that the interaction of anti-poverty programs and cost of living in relation to hippocampal volume was specific to low-income participants who would benefit from anti-poverty programs. We also conducted numerous sensitivity analyses to ensure that alternative state-level social,

**a**

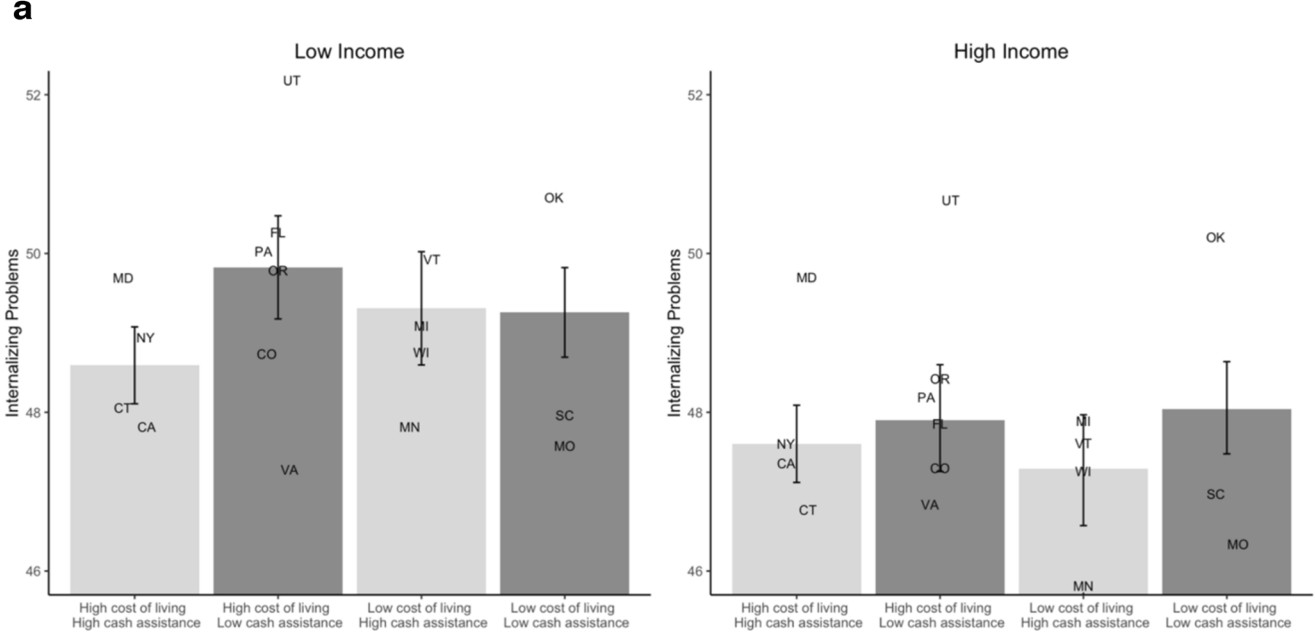

**b**

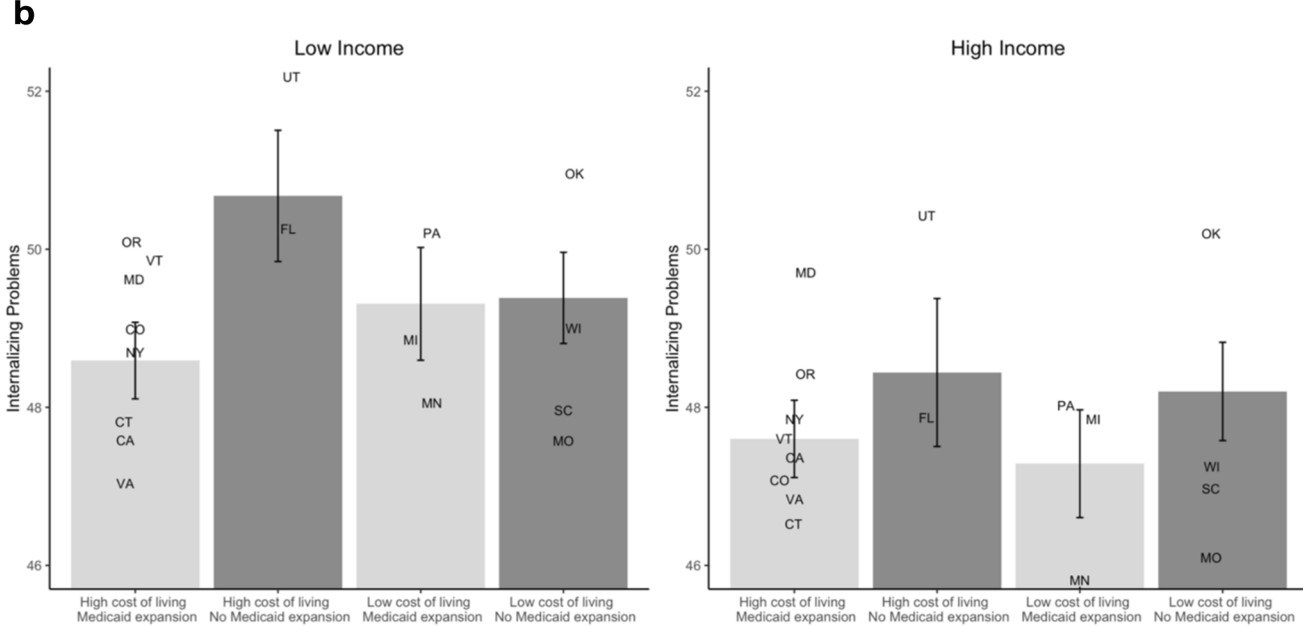

**Fig. 4 | The 3-way interactions among log-income-to-needs ratio, cost of living, and anti-poverty programs in relation to internalizing problems (_N_ = 10,633).**
**a** The simple slopes are plotted to reflect the values actually present in the distribution of cost of living and cash benefits in the ABCD data. The internalizing problems estimate for low cost of living, low cash benefit states is based on the intercept estimates for internalizing problems when log-income-to-needs ratio is 1 SD above or below the mean, and cost of living and cash benefits are both 1 SD below the mean. The estimate for high cost of living, high cash benefit states is based on the intercept estimates when cost of living and cash benefits are both 1 SD above the mean. The estimate for high cost of living, low cash benefit states is based on the intercept estimates when cost of cost of living is at the mean, and cash benefits are 1 SD below the mean, as no state with cost of living 1 SD above the mean had low cash benefits in our data. The estimate for low cost of living, high cash benefit states is based on the intercept estimates when cost of living is 1 SD below

the mean, and cash benefits are at the mean, as no state with low cost of living had cash benefits that were 1 SD above the mean in our data. **b** The internalizing problems estimate for states that did not expand Medicaid is based on the intercept estimates for internalizing problems when log-income-to-needs ratio is 1 SD above or below the mean, and cost of living is 1 SD below the mean. The estimates for high cost of living states that expanded Medicaid are based on the intercept estimates when cost of living is 1 SD above the mean. The estimate for high cost of living states that did not expand Medicaid is based on the intercept estimates when cost of cost of living is at the mean, as no states with income 1 SD above the mean did not expand Medicaid. This is illustrated in supplemental Fig. S2. Error bars reflect the 95% confidence intervals of these estimates. The random intercepts for each state when log-income-to-needs ratio is 1 SD above or below mean are represented by that state's abbreviation.

economic, educational, and political characteristics did not explain these patterns. Further, the observational associations we document based on existing anti-poverty programs complement emerging causal evidence of the impact of cash assistance on children's neurodevelopment[28]. That said, we cannot entirely rule out the possibility that other factors beyond cost of living and the generosity of anti-poverty programs may be explaining the variation in associations of family income with hippocampal volume and internalizing problems. Natural experiments may one day be feasible with this dataset as policy changes at the state-level unfold in the future. However, this type of design is not possible at this time with the ABCD data because there have been no changes in these poverty-relevant policies during the data collection period in the states where ABCD data were collected. Nor to our knowledge are there any other datasets that have collected harmonized neuroimaging data on children living in different contexts over time to allow policy changes to be examined as predictors of changes in neural outcomes.

Second, the ABCD study sites are located in only 17 of the 50 states. This restricted range would have reduced our statistical power to detect moderation, and thus our estimates are likely conservative. At the same time, cost of living and the generosity of anti-poverty programs are correlated within those states, which limits our ability to tease apart these factors. Third, the state-level macro-structural characteristics examined here are confounded with scanner and the demographics of the sample at each site to some degree. However, we control for both scanner and the racial and ethnic makeup of the sample in our analyses. While it is possible that scanner differences contribute to systematic differences in volume estimation of the hippocampus (i.e., the mean volume in each state), it is implausible that scanner differences would produce systematic changes in the association between family income and hippocampal volume.

In addition, state-level cost of living is used, which ignores potential variability in cost of living within states. Further, the ABCD sites may vary in the extent to which their local cost of living is consistent with the cost of living in the state. In general, however, the costs of living in the metropolitan areas where the data were collected tracks closely with the costs of living in the states where the data were collected ($r = 0.73$ across the 21 sites). Each ABCD site was a regional center recruiting between 300 and 1000 participants from wide catchment areas that in most cases spanned beyond the metropolitan statistical area in which the site was located. Given that the anti-poverty benefits that are the focus of the paper are policies enacted at the state level, not metropolitan area, using cost of living estimates from the state-level ensures that the fixed effects in the model are drawn from the same geographical level of analysis.

Another potential issue is that the parent-reported family income could include cash assistance received by the family. Consequently, the income measure may already include variance related to the putative moderator. However, when repeating analyses using a dichotomous indicator of poverty, the results were consistent with the current findings. Further, more generous anti-poverty programs, particularly cash assistance, may not reduce stress by increasing monthly or yearly income in all cases. For example, these programs may allow families to make decisions that lead to a decrease in wages but that also reduce stress, such as working fewer hours.

Finally, all participants in the sample were 9–11 years old at the time of data collection, so our ability to generalize these findings to other ages is limited. However, as the ABCD cohort ages, and, potentially, as state-level policies and macro-economic conditions change over time, there will be opportunities to address this limitation and expand upon these findings in the future.

Lower family income is associated with smaller hippocampal volume and higher internalizing problems. However, we show that these relationships are moderated by state-level macrostructural characteristics. Associations of income with hippocampal volume and internalizing symptoms are magnified in states with high cost of living but reduced in states with more generous anti-poverty programs. Socioeconomic disparities in hippocampal volume and internalizing problems were reduced by 34%–48%, respectively, in high cost of living states with more generous anti-poverty programs vs. states with less generous programs. In states where anti-poverty programs were more generous, the associations of family income with hippocampal volume and internalizing problems in high cost of living states resembled that of low cost of living states. This finding demonstrates that macro-structural conditions related to poverty are associated with the strength of the association between family income and children's neurodevelopment and mental health and suggests that public policies that increase family's financial resources are relevant for efforts to reduce socioeconomic disparities at both neural and behavioral levels. Investments in social safety net programs may therefore contribute to considerable long-term financial savings, given the high cost of addressing mental health, educational, and economic challenges resulting from socioeconomic disparities in neurodevelopment.

## Methods
### Sample
Data were obtained from the ABCD curated data release 3.0 (https://abcdstudy.orghttps://abcdstudy.org) from the NIMH Data Archive using the NIMH Data Archive Download Manager. Informed written consent for child and parent was obtained from parents, and child participants separately completed a written assent. All protocols were performed in accordance with the ethical standards as outlined in the 1964 Declaration of Helsinki and approved by the central Institutional Review Board (IRB) at the University of California, San Diego and/or local IRBs at the individual study sites. We drew data from the baseline assessment of 11,864 youth with data on parent-reported psychopathology and 11,533 youth with brain structure data (5291 female, 47.9%, mean age 9.91 years, SD = 0.622). Twenty-one study sites from 17 states were included from across the U.S. From these sites, a stratified probability sample of schools within the catchment areas for each site were selected, and eligible youth were recruited from each school. The ABCD study approximates a multi-stage probability sample but is not nationally representative (for greater details on sample selection see ref. 38). The imaging procedures were harmonized across sites[39].

### Measures
**Family income.** Parents reported total family income at the baseline visit by selecting one of 10 income ranges. The midpoint of the income range selected was used for each participant. Parents also reported the number of people in the household. Income-to-needs ratio was calculated by dividing the parent-reported family income by the poverty threshold for a family of that size for the year 2017 as indicated by the U.S. Census Bureau[40]. Consistent with prior work on childhood socioeconomic status (SES) and neurodevelopment[5,41], the natural log of income-to-needs ratio was used as a measure of family income in all analyses to reflect that associations of income with neural outcomes are stronger at the lower end of the income distribution (Fig. 1). Site-level differences in the average number of family members living in the home are small, and there is no evidence that site-level variability in family size is likely to be a confounder in these analyses. The distribution of family income-to-needs ratios within each of the 17 States in the ABCD sample are summarized in supplemental Table S7.

**Hippocampal volume.** Hippocampal volume at the baseline visit was obtained from the structural data release. This data release had volume estimates for 11,533 participants. ABCD guidelines were followed with regard to exclusion of participants based on data quality, including exclusion of structural data that was rated as severe in any of the five

categories of image artifact or reconstruction inaccuracy: motion, intensity inhomogeneity, white matter underestimation, pial over-estimation, and magnetic susceptibility artifact[42]. A total of 471 participants were excluded from further analyses based on these criteria. The rate of exclusion varied between study sites (Supplemental Table S8). However, the rate of exclusion was unrelated to cost of living or the generosity of anti-poverty programs.

Volume measures of left and right hippocampus, obtained using automatic segmentation in FreeSurfer 5.3, were summed to produce a measure of total hippocampal volume to reduce the number of analyses and because there is no apparent pattern of laterality in the association between SES and hippocampal volume in the literature[7,11–15]. Automatic segmentation was also used to obtain estimates of total intracranial volume, which was controlled for in all analyses on hippocampal volume.

**Internalizing and externalizing problems.** Internalizing and externalizing problems were assessed using parent reports on the Child Behavior Checklist (CBCL)[43], collected at the baseline visit. The age-corrected T-scores were used for each measure.

**State-level moderators.** The following state-level moderators characterize the U.S. state of the site in which each participants' data was collected. These state-level characteristics as well as the demographics of the sample at the state level are summarized in Table 1.

*Cost of living* is based on Regional Price Parity (RPP) for the year 2017—the median year in which the ABCD baseline data was collected—obtained from the U.S. Bureau of Economic Analysis[21]. RPP indexes the expected cost of an equivalent item in each state as a factor of the mean cost across all 50 states. For example, an item that costs $1 on average costs $1.10 in California (RPP = 1.10) and 92 cents in Oklahoma (RPP = 0.92).

*Cash assistance* is the mean of the average monthly Earned Income Tax Credit (EITC) and Temporary Assistance for Needy Families (TANF) benefit in each state. States vary in the amount of additional EITC they grant as a proportion of the federal EITC. In 2017, the average federal EITC for families with children was $266 a month[44]. The monthly EITC benefit in each state was therefore calculated as $266 x (1 + state EITC as a proportion of federal EITC). The mean was then taken between the average monthly EITC (range: $266-$399) and the monthly TANF benefit[45] (range: $286–$789) in each state to compute a variable indicating the average cash benefit received by either unemployed (TANF) or working (EITC) low-income families in that state.

*Medicaid expansion* is a dichotomous variable indicating whether or not that state had expanded Medicaid eligibility through the Affordable Care Act by the end of 2017[46].

**Analysis**
Bivariate correlations between all study variables are provided in Supplemental Table S9. Analyses were conducted using linear mixed-effects models with the nlme package in R using two-tailed tests. For all analyses, the assumptions of homoscedasticity and independence were confirmed by plotting and visually inspecting model residuals. Normality was confirmed using the Wilk-Shapiro test (all $W < 0.96$, all $p < 0.001$). Analyses were restricted to participants who reported family income. These included 9913 participants for analyses with hippocampal volume as the outcome and 10,633 participants for analyses with mental health outcomes. The proportion of participants missing data on income varied between states but was unrelated to cost of living or the generosity of anti-poverty programs. Random intercepts were included for study site. While including random effects for family proved untenable, as the models all failed to converge, we reran all analyses, randomly including only one sibling from each family with multiple siblings in the sample, and results remained

unchanged (see Supplemental Table S10). To estimate the association between income-to-needs ratio and hippocampal volume, fixed effects were included for log-income-to-needs ratio, total intracranial volume (ICV), scanner, age, and sex. To determine the association between log-income-to-needs ratio and mental health (internalizing problems, externalizing problems), fixed effects were calculated for income-to-needs ratio, age, and sex.

Cost of living and cash assistance were grand mean centered. To evaluate state-level moderation, random intercepts and random slopes of income-to-needs ratio at the study site level were included in all analyses. To determine if the association between income-to-needs ratio and each outcome was moderated by cost of living, cash assistance, or Medicaid expansion, a separate model was fit for each state-level moderator in which fixed effects were included for income-to-needs ratio, the proportion of the sample at that study site that was White, Black, and Latinx, anti-poverty programs, each of the 2-way interactions between anti-poverty programs, cost of living, and log-income-to-needs ratio, and the 3-way interaction among log-income-to-needs ratio, cost of living, and each of the anti-poverty programs. In models where hippocampal volume was the outcome, fixed effects were also included for ICV and the type of MRI scanner used at each site. Models were also fit with only the 2-way interactions between cost of living and family income and anti-poverty programs and family income, without the 3-way interaction, or the interaction between cost of living and anti-poverty programs. Results of those models are provided in Supplemental Table S1.

To evaluate whether anti-poverty programs were plausible moderators of the association between family income and these outcomes, we examined whether model fit was improved when random slopes were included in the model. Based on the likelihood ratio test, model fit was significantly improved when random slopes were included for the relation between log-income-to-needs ratio and externalizing problems (likelihood ratio = 53.05, $p < 0.001$), but not for hippocampal volume (likelihood ratio = 3.36, $p = 0.186$) and internalizing problems (likelihood ratio = 5.72, $p = 0.057$) when random effects were included in the model at the site level. However, when a model was instead fit with clustering at the *state* level, inclusion of random slopes significantly improved model fit for both hippocampal volume (likelihood ratio = 8.06, $p = 0.018$) and internalizing problems (likelihood ratio = 7.86, $p = 0.020$), suggesting that state-level anti-poverty programs are plausible moderators of the association between log-income-to-needs ratio and these outcomes. Importantly, significant cross-level interactions are possible even when inclusion of random slopes does not significantly improve model fit in the types of multi-level models estimated here[47].

**Sensitivity analyses**
We conducted sensitivity analyses controlling for 10 state-level social, economic, educational, and political characteristics that may serve as alternative explanations for state-level variability in hippocampal volume, internalizing problems, and their associations with SES. The control measures are described and rationales for their inclusion are provided in the supplemental materials. The results of the sensitivity analyses are summarized in Supplemental Tables S3 and S4.

We also conducted sensitivity analyses to address some other issues. We included the percentage of participants at each site that were Black, White, and Latinx as covariates in all analyses, as site-level demographics are plausible confounders. Conversely, it is our view that using individual-level racial categories as variables of interest or covariates presumes a biological basis for these racial categories that is not supported by evidence. However, in the interest of demonstrating that our results hold whether race and ethnicity are included as site-level or individual-level variables, we conducted additional sensitivity analyses controlling for race and ethnicity at the individual level, and our results remain virtually unchanged (Supplemental Table S11).

**Reporting summary**

Further information on research design is available in the Nature Portfolio Reporting Summary linked to this article.

## Data availability

Data is from the Adolescent Brain and Cognitive Development Study. Information on how to access ABCD data through the NIMH Data Archive (NDA) is available on the ABCD study data sharing webpage: https://abcdstudy.org/scientists_data_sharing.html. https://abcdstudy.org/scientists_data_sharing.html. Deidentified data for the current analyses can be found at https://osf.io/t3ev7/.

## Code availability

All code for aggregating data from files obtained from the NIH Data Archive as well as all analytic code can be found at https://osf.io/t3ev7/.

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

## Acknowledgements
Support for this research came from R01-MH106482, R56-MH119194, and R37-MH119194 to KM, K99-MH127248 to D.W. from the National Institute of Mental Health, and National Science Foundation CAREER award BCS-1653188 to M.C. The funders had no role in study design, data collection and analysis, decision to publish, or preparation of the manuscript. The authors have no financial disclosures or conflicts of interest to declare. We are grateful to Marianne Page who provided advice during the initial conceptualization of this study and to John Flournoy who provided feedback on the manuscript and analytic approach.

## Author contributions
D.G.W., M.L.H., M.C., and K.A.M. contributed to the development of the research questions and analytic strategy. D.G.W. conducted all analyses and wrote the manuscript. D.M.B. conceptualized the broader study design and contributed essential information for the analytic strategy. All authors reviewed and provided essential feedback, edits, and revisions on the manuscript.

## Competing interests
The authors declare no competing interests.
