## [Peer Review File · Nature Communications]

State-level macro-economic factors moderate the association of low income with brain structure and mental health in U.S. childrenReviewers' comments:

Reviewer #1 (Remarks to the Author):

This paper focuses on a centrally important question in terms of understanding socioeconomic disparities, the degree to which association with hippocampal volume and mental health are reduced in the context of stronger antipoverty policies and benefits. The approach is to utilize the multi-site ABCD study, and significant moderation analyses are used to postulate that strengthening structural antipoverty interventions are a promising strategy to reduce disparities. While this is an important question that is relevant to both theory and applied policy decision to promote equity, and thus is both interesting and significant question/approach, there are a number of concerns that need to be addressed to support the conclusions.

First, from a theoretical level it would be important to clarify if the authors are focused on antipoverty programs as the specific instrument by which such moderation (or policy impacts would occur), or if antipoverty programs are being used as a marker for a broader set of macro-level conditions within the state. Although in general the majority of the paper is written as the former (that is, that antipoverty programs will reduce SES-related disparities), and the focus on identifying specificity by ruling out alternative indices of state-level characteristics, there are aspects of the methodology and discussion that suggests the latter. Given that most of the paper is focused on the former, it is important to also provide a stronger justification for assuming that the state-level policy is a strong and reliable marker for the level of uptake of the benefits, and that this assumption would hold similarly across states. The integration of state-level data with individual-level data is a strength, but again presumably the argument is individual families will benefit from those policies, and additional state-level data justifying the invariance of this assumption would make a stronger case.

An additional theoretical and methodological challenge is that presumably one mechanism for cash assistance policies as a protective factor is that they functionally increase income or resources, which could reduce stress but also impact other mechanisms as well. The challenge here, then, is in the income measure itself, as it appears that what is being used is the total, combined family income measure. As worded, that item in ABCD asks caregivers to include all sources of income in their answer, including income from various benefit programs such as unemployment benefits. Consequently, the income measure may already include variance related to the putative moderator, and thus it is challenging to conceptualize how such moderation would occur if it is already incorporated in the income measure. There are other income measures that could be used that do not include such language, including specific questions around income for the primary caregiver and partner that could be combined to avoid this problem.

Moreover, while utilizing the natural log of income-to-needs is a typical approach to income, it does not appear to fit the important question the authors pose that appears specifically focused on the role of anti-poverty programs. In particular, keeping such a continuous measure means that much of the association being modeled is for the association between income and outcomes for those who would not be eligible for support from the anti-poverty program or would not be impacted directly by Medicaid expansion in terms of eligibility for insurance (such as for those perhaps for an income-to-needs of 2 or higher), which in this case would appear to be the majority of the sample. Given the role of antipoverty programs themselves does seem to be the focus, it would be much stronger, and much more directly relevant, to operationalize this variable in terms of those who would be eligible for support from benefits (possibly in general or in a state-specific manner), to identify if generosity of programs reduces the differences between those who are potentially eligible or not eligible for support. One could even follow-up by creating three categories to separate those who are not eligible for support but experience high economic need, as this population may be especially vulnerable in high cost of living states.

The authors include a laudable focus on variation by site, not only in terms of anti-poverty programs

but in terms of a general consideration of variation by macro-level contexts. This would be much clearer and stronger with some additional steps. First, please provide the distribution of income by site/state, to help clarify the role within-site variation in income might have in understanding moderation. Second, please clarify the inclusion/exclusion steps by which participants are dropped from analyses (and the sample size for those dropped at each step), and examine whether there is differential inclusion/exclusion by site. Third, prior to moderation, it is important to strengthen the consideration of variation in slopes/intercepts by site. Right now it seems solely focused providing the standard deviation, and it isn't clear if that is the standard deviation in parameter estimates by site. Nevertheless, it is important to not only show a standard deviation, but determine if the variation in parameters for income differ by site, and thus a significance test is needed to determine whether what is reported as variance is truly indicative of variation by site and not simply expected noise around an overall parameter mean. Finally, it would be helpful to clarify if the interactions are modeled such that they are actually cross-level interactions between site-level variables (thus accounting for clustering) and income at the individual family level, in order to represent the hypotheses accurately. Finally, please run sensitivity analyses without sites from CA and FL, to determine if results are robust to omitting cases where there are multiple sites per state (give the issue, that should be discussed, in which state-level data is applied to sites though they may differ in social and economic characteristics from the state as a whole).

Some additional questions arise from the moderation analyses, which as the authors note is could be limited the state-level sample size (17). In particular, when considering simply the intersection between cost-of-living and generosity of support, it appears (especially from Figure 2) that there is very minimal representation of (a) low cost of living but generous programs (0 or 1 state, and (b) for Medicaid in particular, high cost of living but expanded support (this is also true for cash assistance at the higher end for cost of living). This raises concerns about the stability of this interaction, and whether it would be stronger instead for focus primarily on two-way interactions, either for each variable separately, or for a combined variable that represents the primary categories. In addition, it wasn't clear whether the two-way interaction model was tested first, and then the three-way interaction. Given the potential challenge with the distribution of categories, it would help to build this model sequentially and determine whether adding the three-way interaction actually improves the model fit over the two-way interaction models. For all such moderation analyses, it is also important to report on all the main effects and interactions in the model so that the overall pattern can be interpreted, rather than only the significant parameters.

In addition, with respect to analyses, it is important to account for family-level nesting within sites in ABCD, which was not mentioned. This could be done either through random exclusion of family members or accounting for family-level clustering in the mixed modeling approach. Additionally, it is important to control for race/ethnicity, particularly given that this variable is included at the site level. Moreover, it was surprising not to see parental education utilized as a covariate as well, or an affirmative and strong argument made for its exclusion.

Additional

- The timing of the data used in the longitudinal study was not clear. It says "We drew data from the Year 1 assessment", but then much data actually appears to be from baseline though this is not always specified. The authors should separate the data release version from the year of assessment, clarify which measures come from which time points in the study and why, and thus which analyses are cross-sectional and which are longitudinal.
- The focus of the title on "brain structure" suggests a wider or more pervasive, global pattern. It would be better to focus more specifically on hippocampal volume, as it is the only aspect of the brain that was investigated.
- Additional papers have been published using ABCD that examine how effects vary by site, and thus the paragraph in the discussion of the methodological innovation should review this literature and place the current paper in context.

Reviewer #2 (Remarks to the Author):

In this study, ABCD study data was utilised to address the question of whether anti-poverty policies/cash benefits for lower SES families reduced the association between SES and hippocampal volume/internalizing/externalizing symptoms in children. Results were supportive of hypotheses. This is a very interesting study with important implications. I have a few comments/suggestions for changes:

1. Abstract: Please specify that the sample is from the US. Throughout: it should be made clear that much of the discussion is, or may be, only relevant to the US population. Some US-specific things are discussed but not explained, e.g., Medicaid
2. Introduction: Some statements are overly speculative, e.g., "Increasing evidence demonstrates that SES can influence the developing brain" and "Childhood SES has consistently been found to influence brain structure". Does the evidence point to SES having a causal influence on the brain? My understanding is that the existing studies descriptive/correlational.
3. "This finding suggests that reductions in stress exposure may be one pathway through which state-level anti-poverty programs exert a buffering effect on hippocampal volume and internalizing problems in high cost of living states." This may be true, but the authors have not tested the relevant (mediation) model to address this question. I understand that mediation is not ideal for cross-sectional data, but if there is a theory there, then I think a moderated mediation model would be useful to provide support for the authors' claim (with appropriate acknowledgement of the limitations of cross-sectional data).
4. What is the rationale for using data from the year 1 assessment only? Is this the 1-year follow-up?

Reviewer #3 (Remarks to the Author):

Review of Weissman et al.: Antipoverty programs mitigate socioeconomic disparities in brain structure and psychopathology among U.S. youths

In this report, the authors set out to assess whether public policies that increase financial resources for families with low income can reduce socioeconomic disparities in brain development and mental health using the ABCD cohort which spans 21 sites across 17 states in the US. They operationalized antipoverty programs at the state level, apparently essentially assuming that every family within each of the 17 states were availed of monthly Earned Income Tax Credit (EITC; \$266-\$399/month) and/or Temporary Assistance for Needy families (TANF \$286-\$789/month), with some states being more generous in financial assistance than others. They hypothesized lower SES to be associated with smaller hippocampal volume, higher internalizing and externalizing problems, and greater exposure to stressful life events, and that antipoverty programs would mitigate and decrease these associations, particularly in high cost of living states. Based on statistical analyses to test these hypotheses, the authors conclude that the magnitude of previously observed associations between brain and SES did vary as a function of state-level macro-economic factors—including the cost of living and generosity of anti-poverty programs, consistent with their hypotheses. While findings of this nature could have significant implications for public policy in improving the lives of disadvantaged children, there are numerous issues with the analyses and interpretation of results that indicate further investigation is required to draw such conclusions.

Specific comments:

1. The primary concern is that when using metrics at the state level, assumptions are made about similarity in demographics among the various 21 ABCD cohorts that are not supportable. While the ABCD cohort as a whole largely reflects the demographics of the US population, at the site level, they do not. For example, there are 4 distinct sites in California, 2 in Los Angeles (USC/CHLA and UCLA), 1 in the Stanford area of northern California, and one in San Diego. The CHLA site has proportionally the largest number of very low-income families, while, the Stanford site has proportionally the largest number of very high income families in the ABCD cohort. Both Los Angeles and Stanford have a very high cost of living, equating the populations within the ABCD cohort at those 2 sites would be misleading. While the cost of living is undeniably lower in cities like Tulsa, Oklahoma and Salt Lake City, Utah than the California sites, the proportion of underrepresented minorities is also much lower at those 2 sites. While it is true that some metrics of population characteristics at the state-level could be meaningful and informative, additional family income from antipoverty programs considered as equal in Stanford and Los Angeles are probably not.
2. The conceptual significance of the results are conveyed as the percent difference in the relationship between brain volume and income to needs with and without accounting for antipoverty benefits (I think?). For example, the authors report that "among adolescents living in states with higher cost of living, the association between family income and hippocampal volume was about 37% smaller in states with more generous cash assistance programs compared to states with less generous cash assistance programs." The problems with this statement include that it seems somehow like they are saying that the hippocampus is 37% smaller, but they are talking about the statistical correlation (I think), which leaves me wondering 37% of what?
3. In reading the report, I keep thinking that adding government assistance from antipoverty measures could be just like additional family income. With that higher income, of course the oft-observed relationship between family income and brain volume would be attenuated because the income is essentially higher with higher antipoverty cash benefits. Further, equating additional income for all participants in each state within the ABCD cohort doesn't make sense either because we know from previous reports within the whole ABCD cohort that additional increased increments of income make a much bigger impact on SES brain volume relationships in the lower income than higher income families. So, for example, an additional \$600/month would make the difference between paying rent or buying food for a family making \$50k/year but might pay for a summer vacation for a family making \$150k/year (see Gonzalez et al., 2020, Frontiers in Human Neuroscience Figure 1 for relationships between income to needs and total brain volume and total cognition scores in the entire ABCD baseline cohort).
4. There are also site level differences in the number of family members living in the home which could make income-to-needs a bit skewed, I think. The authors do not mention anything in the methods or supplementary figures about how they are dealing with multiple siblings or even the twin cohorts in their analyses. Don't families with multiple children get government assistance for each child? So, if a family has 2 children, they would get twice the monetary assistance of a family with only one child?
5. It is not at all clear why the authors account for race differences at each site as the proportion of the sample at that study site that was White, Black, and Latinx, rather than using that variable at the individual level of participant self-report of race/ethnicity.

Minor concerns:

6. On multiple occasions in the manuscript, the authors use the term Adolescent Behavior and Cognitive Development (ABCD) study.
7. Using just one region of interest, hippocampal volume (statistically corrected for ICV) does not show any specificity to that particular region.
8. There are many geocoded measures at the census tract level that could be used to estimate cost of living at the individual participants home address rather than having a single cost of living measure collapsed across all participants who live in each state.

Reviewer #4 (Remarks to the Author):

Summary: This study uses national data from the multisite ABCD study to examine the association between individual-level and state-level socioeconomic factors with brain development and mental health. ABCD is a rich data set, and evaluating the effects of anti-poverty policies is critically important. The manuscript is also well written and the figures are clear and well labeled. However, the fundamental flaw of this manuscript is that the correlational design of the current analysis is overly simplistic, making it not particularly rigorous or well-suited to address the multitudinous sources of confounding. That is, simply regressing each outcome on the exposure of interest while adjusting for various covariates does not provide strong support for the hypothesis that these exposures matter for these outcomes. Here are additional details on my methodological concerns:

1. The individual-level SES variable was simply constructed using just income and household size. The exposure-outcome relationship could be confounded by any number of unmeasured SES or other confounders, including experiences of racism, wealth, housing, immigration status, etc. There is a rich literature in social epidemiology about the importance of these many operationalizations of SES.
2. Similarly, the state-level variables are a small selection of policies that are likely to be confounded by many other characteristics of states that are likely to affect both the exposure and outcome. The University of Kentucky Center for Poverty Research has a rich database of other types of state characteristics and policies that are commonly included as covariates in studies like this: <https://ukcpr.org/resources/national-welfare-data>
3. Furthermore, the state variables are ecological in nature, i.e., state-level averages of typical benefit levels. There is no attempt to restrict the sample to those individuals for whom cash assistance or safety net policies matter most, e.g., racial/ethnic marginalized communities, low-income individuals, etc. Indeed, it would be worthwhile to test whether there is an association between these exposures/outcomes among high-income groups, which, if found, would support the idea that the relationship is strongly confounded.
4. On that note, there is no table of sample characteristics to indicate the degree of socioeconomic vulnerability in this sample. What percent of the sample is low-income, what is the racial/ethnic make-up, how old are the children, what is the educational level of their parents, etc., etc.?
5. Even if the issues above were addressed, fundamentally this would still be a correlational analysis that does not contribute much new information to the existing literature on poverty and health. This analysis would be strengthened by more rigorously evaluating a single policy using a quasi-experimental technique, as has been done in prior research on the EITC, WIC, etc. E.g., Strully et al, ASR 2010; Hoynes et al JPE 2011.
6. Alternately, this study could take advantage of the fact that the ABCD study includes multiple waves. The current study only used Wave 1. Using multiple waves and employing an individual-level fixed effects design would strengthen causal inference about the effects of individual-level factors, or a state-level fixed effects design could provide stronger evidence about the effects of time-varying state-level factors. E.g., see Brüderl and Ludwig. "Fixed-effects panel regression." The Sage handbook of regression analysis and causal inference (2015).
7. It's unlikely that something like Medicaid expansion could so quickly act to change hippocampal volume, i.e., there is not a plausible mechanism for such a rapid effect. In fact, this finding again suggests that some other source of confounding at the state level is present.

We have included a revised manuscript. A detailed response to each of the reviewer comments is below in **bold**.

Reviewer #1 (Remarks to the Author):

This paper focuses on a centrally important question in terms of understanding socioeconomic disparities, the degree to which association with hippocampal volume and mental health are reduced in the context of stronger antipoverty policies and benefits. The approach is to utilize the multi-site ABCD study, and significant moderation analyses are used to postulate that strengthening structural antipoverty interventions are a promising strategy to reduce disparities. While this is an important question that is relevant to both theory and applied policy decision to promote equity, and thus is both interesting and significant question/approach, there are a number of concerns that need to be addressed to support the conclusions.

First, from a theoretical level it would be important to clarify if the authors are focused on antipoverty programs as the specific instrument by which such moderation (or policy impacts would occur), or if antipoverty programs are being used as a marker for a broader set of macro-level conditions within the state. Although in general the majority of the paper is written as the former (that is, that antipoverty programs will reduce SES-related disparities), and the focus on identifying specificity by ruling out alternative indices of state-level characteristics, there are aspects of the methodology and discussion that suggests the latter. Given that most of the paper is focused on the former, it is important to also provide a stronger justification for assuming that the state-level policy is a strong and reliable marker for the level of uptake of the benefits, and that this assumption would hold similarly across states. The integration of state-level data with individual-level data is a strength, but again presumably the argument is individual families will benefit from those policies, and additional state-level data justifying the invariance of this assumption would make a stronger case.

Response: We thank the reviewer for this important point and want to clarify that we are using the policies selected for analysis as indicators of the broader structural climate for families with low income. The macrostructural characteristics examined in our paper are intertwined with demographic, historical, economic, and political characteristics that shape regional differences in the structural environments that low-income families navigate. In this revision, we have been careful to communicate more clearly that this paper does not serve as a direct evaluation of specific policies, which would require a natural experiment or randomized controlled trial. However, the variables selected in this study serve as quantitative indicators of the broader structural environment in which families with different levels of income are embedded, which we are able to operationalize on a meaningful scale (dollars), and at a level of regional variability (U.S. states) that is both meaningful in the real world (i.e., it is the geographic unit at which these policies are implemented) and aligned with the sampling procedures of the ABCD study.

As the reviewer astutely points out, it would be unjustified to assume that eligibility and uptake of cash assistance is homogenous across states. Indeed, states with less generous benefits also often have stricter eligibility requirements and lower uptake of benefits. This fact is captured in part by the TANF-to-poverty ratio (TPR): the number of families receiving TANF benefits for every 100 families with children in poverty. State TPRs in the ABCD states in 2017 ranged from 8% in Oklahoma to 68% in California (Shrivastava & Azito Thompson, 2022). Importantly, state TPR is strongly

correlated with the mean cash assistance estimates used in our analyses at $r=.76$. Therefore, while the cash assistance estimates are quite conservative estimates of the disparities in the assistance families might realistically expect to receive in any given state, they nonetheless are consistent reflections of the rank order of the overall generosity of antipoverty programs in that state. And as we articulate in the revision, the generosity of anti-poverty policies is an important and quantifiable indicator of the social safety net for families living in poverty that varies meaningfully across U.S. states.

An example of language we have added to make the goal of our analyses clear is provided on p. 6 of the revised manuscript:

“Importantly, the correlational analyses described in this study are not intended to serve as evaluations of these specific policies. Rather, the policies selected in this study are intended to serve as quantitative indicators of the broader structural environment in which families with different levels of income are embedded, which we are able to operationalize on a meaningful scale (e.g., dollars) and at a meaningful level of regional variability (U.S. states).”

We additionally address this concern with new analysis. Specifically, we replicate our analyses using an additional state-level indicator of the generosity of the social safety net in each state for families with low income - the state-level minimum wage. As now described on p. 10 and p. 11 of the manuscript, in supplemental analyses, we examined the 3-way interactions of income, cost of living, and the state minimum wage in relation to hippocampal volume, internalizing problems, externalizing problems, and found the interaction was significant in all analyses (see supplemental table S7 and S8). This also now informs our interpretation on p. 14:

“Given that we observed the same type of 3-way interaction of cost of living and family income with cash assistance, Medicaid expansion, we expect that these policy differences—while helpful in and of themselves—are likely indicative of a broader collection of policies and structural supports for families facing economic hardship that were not directly measured in this study (e.g., availability of free or reduced cost early childhood education programs). Indeed, in supplemental analyses, we examined the 3-way interactions of income, cost of living, and the state minimum wage in relation to hippocampal volume, internalizing problems, and externalizing problems, and found the interaction was significant in all 3 analyses.”

An additional theoretical and methodological challenge is that presumably one mechanism for cash assistance policies as a protective factor is that they functionally increase income or resources, which could reduce stress but also impact other mechanisms as well. The challenge here, then, is in the income measure itself, as it appears that what is being used is the total, combined family income measure. As worded, that item in ABCD asks caregivers to include all sources of income in their answer, including income from various benefit programs such as unemployment benefits. Consequently, the income measure may already include variance related to the putative moderator, and thus it is challenging to conceptualize how such moderation would occur if it is already incorporated in the income measure. There are other income measures that could be used that do not include such language, including specific questions around income for the primary caregiver and partner that could be combined to avoid this problem.

Response: The Reviewer is correct that the total family income variable could include cash assistance received by the family. However, the method of reporting income in the ABCD study is not sufficiently precise for this to make a substantial difference in the total income reported by each caregiver. The bins used to report income at the lower end of the distribution are: <\$5,000; \$5,000-\$11,999; \$12,000-\$15,999; \$16,000-\$24,999; \$25,000-\$34,999; and \$35,000-\$49,999. These bins are

sufficiently large that the antipoverty benefits received, while likely helpful, are in most cases not sufficient to move a family from one bin to the next.

To address this comment more directly, we also performed the analyses suggested by the Reviewer using the sum of parent and partner income as an alternative measure of family income. Family income, when calculated using the total combined income item used in our original analysis, is correlated with the sum of parent and partner income at $r=.90$. Estimates from summing parent and partner income as suggested by the Reviewer, which should in theory be lower than the total income variable we used as they do not include any antipoverty benefits, are actually significantly higher than the family income reported using the combined item (Mean difference = \$9,755, $p<.001$). This is true in the entire sample as well as for both lower (income-to-needs ratio < 2 , Mean difference = \$3,135, $p<.001$) and higher income participants (income-to-needs ratio > 2 , Mean difference = \$12,928, $p<.001$). There is therefore no suggestion in the data that the total income estimates as reported are influenced by the receipt of antipoverty benefits in a way that would change our results.

Finally, more generous antipoverty programs (particularly cash assistance) may not reduce stress by increasing in monthly or yearly income in all cases. For example, they may allow families to make decisions that lead to a *decrease* in wages but that also reduce stress, such as working fewer hours.

Moreover, while utilizing the natural log of income-to-needs is a typical approach to income, it does not appear to fit the important question the authors pose that appears specifically focused on the role of antipoverty programs. In particular, keeping such a continuous measure means that much of the association being modeled is for the association between income and outcomes for those who would not be eligible for support from the anti-poverty program or would not be impacted directly by Medicaid expansion in terms of eligibility for insurance (such as for those perhaps for an income-to-needs of 2 or higher), which in this case would appear to be the majority of the sample. Given the role of antipoverty programs themselves does seem to be the focus, it would be much stronger, and much more directly relevant, to operationalize this variable in terms of those who would be eligible for support from benefits (possibly in general or in a state-specific manner), to identify if generosity of programs reduces the differences between those who are potentially eligible or not eligible for support. One could even follow-up by creating three categories to separate those who are not eligible for support but experience high economic need, as this population may be especially vulnerable in high cost of living states.

Response: Thank you for this helpful comment. We agree, as the Reviewer suggests, that cost of living and antipoverty programs should only influence our outcome variables for people in the low-income range who are actually eligible for benefits. It is possible that by using a continuous measure of family income as our independent variable our effects might be driven by participants who could not benefit from antipoverty programs. However, we can provide evidence that it is indeed the participants at lower income levels who are specifically impacted by these macrostructural policy differences.

First, we ran the additional sensitivity analyses suggested by the Reviewer. As shown in the table below, if we recode income dichotomously as participants with lower income who could potentially benefit from these programs (i.e., above vs. below the federal poverty line) vs. those with higher incomes who could not, the results are consistent with our original findings. However, if we use a higher cutoff that does not correspond with eligibility for benefits (i.e., above vs. below 5x the federal poverty line), the 3-way interaction is not significant for any analysis.

Income dichotomized at the poverty line				Income dichotomized at <5x the poverty line			
	B	SE	p		B	SE	p
Poverty	-172.7	28.1	<.001	< 5x Poverty (Low Income)	-99.1	20.7	<.001
Cost of Living (COL)	-136	332	.688	Cost of Living (COL)	223	374	.560
Mean Cash Benefit (Cash)	.071	.241	.771	Mean Cash Benefit (Cash)	-.141	.294	.638
Poverty x COL	-222	377	.547	Low Income x COL	-546	351	.120
Poverty x Cash	.094	.300	.753	Low Income x Cash	.416	.282	.140
COL x Cash	1.22	1.85	.521	COL x Cash	-.83	2.00	.682
Poverty x COL x Cash	4.86	2.14	.023	Low Income x COL x Cash	2.73	1.90	.151
	B	SE	p		B	SE	p
Poverty	-206.9	45.5	<.001	Low Income	-96.9	58.8	.100
COL	-164	501	.747	COL	187	670	.783
Medicaid Expansion (ME)	-3.7	47.9	.939	ME	-5.2	63.6	.935
Poverty x COL	-1029	522	.049	Low Income x COL	-546	654	.404
Poverty x ME	64.8	53.0	.222	Low Income x ME	-3.4	62.4	.956
COL x ME	227.9	539.6	.679	COL x ME	-192	711	.79
Poverty x COL x ME	1396	591	.018	Low Income x COL x ME	874	698	.211
Internalizing Problems							
Income dichotomized at the poverty line				Income dichotomized at <5x the poverty line			
	B	SE	p		B	SE	p
Poverty	2.04	0.53	<.001	Low Income	.966	.443	.029
COL	-2.09	8.76	.815	COL	8.26	11.7	.493
Cash	-.00191	.00692	.787	Cash	-.00783	.00933	.416
Poverty x COL	6.61	7.12	.354	Low Income x COL	-9.59	7.73	.215
Poverty x Cash	-.00717	.00576	.213	Low Income x Cash	.00512	.00627	.415
COL x Cash	.0298	.0558	.601	COL x Cash	.0148	.0705	.837
Poverty x COL x Cash	-.0778	.0414	0.06	Low Income x COL x Cash	-.0057	.0410	.889
	B	SE	p		B	SE	p
Poverty	2.82	1	.005	Low Income	2	1.36	.142
COL	1.15	14.4	.938	COL	5.11	19.8	.834
ME	-.948	1.43	.518	ME	-.41	1.92	.834
Poverty x COL	16.3	11.3	.148	Low Income x COL	2.51	15.0	.867
Poverty x ME	-1.04	1.13	.361	Low Income x ME	-1.2	1.42	.400
COL x ME	-.661	16.2	.968	COL x ME	-4.34	21.4	.843
Poverty x COL x ME	-27.2	12.5	.030	Low Income x COL x ME	-5.73	15.6	.714

Second, we also demonstrate that our effects are being driven by participants with low income by conducting simple slopes analyses breaking down the 3-way interaction between log income-to-needs ratio, cost of living, and generosity of antipoverty programs. These analyses demonstrate that the interactions between cost of living and antipoverty programs are significantly stronger for families with low income. In fact, when hippocampal volume is the outcome, the interaction between cost of living and antipoverty programs is only significant for participants with incomes below the federal poverty line. This is illustrated in Figures 2B and 3B of the revised manuscript (reproduced below). We consider this analytic approach and these results most appropriate to report as they are based on the true range of income variability in the data, including within low-income participants, which is lost when a dichotomous measure of income is used. We now report these simple slopes analyses in the revised manuscript.

The authors include a laudable focus on variation by site, not only in terms of anti-poverty programs but in terms of a general consideration of variation by macro-level contexts. This would be much clearer and stronger with some additional steps. First, please provide the distribution of income by site/state, to help clarify the role within-site variation in income might have in understanding moderation.

Response: We have added this distribution as Table S1 in supplemental materials and reproduced it below. Overall, the distribution in family income-to-needs ratios demonstrates heterogeneity

across states. This heterogeneity roughly corresponded with income distributions in the states from which these samples were drawn, with one meaningful exception. Low-income families were over-represented in the sample from Pennsylvania. Smaller discrepancies were observed in Oregon, Vermont, Minnesota, and Colorado, where the samples were disproportionately higher income compared to the overall state distributions, and in California, Florida, Maryland, and Oklahoma, where the samples were disproportionately lower income than the state, although not nearly to the same degree as the sample from Pennsylvania.

State	Mean Income-to-needs ratio	Below poverty	1-2x poverty	2-5x poverty	>5x poverty
CA	3.76	.20	.14	.32	.34
CO	4.42	.05	.10	.47	.39
CT	4.15	.15	.10	.38	.38
FL	2.55	.29	.22	.35	.14
MD	3.79	.22	.09	.33	.35
MI	3.80	.14	.13	.43	.31
MN	4.49	.03	.10	.46	.40
MO	3.09	.18	.17	.48	.17
NY	3.61	.21	.10	.34	.35
OK	2.72	.24	.22	.37	.17
OR	4.32	.06	.11	.44	.39
PA	1.80	.44	.26	.23	.08
SC	3.71	.16	.15	.36	.33
UT	3.12	.09	.22	.56	.13
VA	3.45	.16	.13	.48	.24
VT	4.70	.04	.08	.39	.49
WI	4.15	.09	.07	.50	.34

Second, please clarify the inclusion/exclusion steps by which participants are dropped from analyses (and the sample size for those dropped at each step), and examine whether there is differential inclusion/exclusion by site.

Response: We now provide additional detail on the criteria for inclusion in each of the analyses and summarize the number of participants excluded for each reason and overall by state in supplemental Table S3. There is no evidence of differential exclusion as a function of income or state based on the quality of the structural MRI data. We did find evidence that states varied in the number of participants who were missing data on income. Those participants were excluded from analyses, as multiple imputation proved untenable for the multi-level models we are testing.

Third, prior to moderation, it is important to strengthen the consideration of variation in slopes/intercepts by site. Right now it seems solely focused providing the standard deviation, and it isn't clear if that is the standard deviation in parameter estimates by site. Nevertheless, it is important to not only show a standard deviation, but determine if the variation in parameters for income differ by site, and thus a significance test is needed to determine whether what is reported as variance is truly indicative of variation by site and not simply expected noise around an overall parameter mean.

Response: First, yes those are the standard deviations in parameter estimates by site. We now clarify that in the manuscript.

Second, conceptually, we agree with the Reviewer that it may not make sense to look for reasons that the associations of income with hippocampal volume and our other outcomes might differ at the state level if they do not in fact differ by site. However, it is important to note that neither the inference criteria nor the power of the likelihood ratio test of model fit for nested multilevel models is equivalent to the statistical test for a cross-level interaction. That is, it is possible for a significant cross-level interaction to occur even when inclusion of random slopes does not significantly improve model fit (LaHuis & Ferguson, 2009).

With that conceptual heuristic and statistical reality in mind, we now provide a more comprehensive description of the variability in the random slopes of log income-to-needs ratio in relation to the three outcomes in this study on p. 21 of the revised manuscript. Based on the likelihood ratio test, model fit was significantly improved when random slopes were included for the relation between log income-to-needs ratio and externalizing problems (likelihood ratio = 54.02, $p < .001$), but not for hippocampal volume (likelihood ratio = 4.27, $p = .119$) and only marginally for internalizing problems (likelihood ratio = 5.77, $p = .056$) when random effects were included in the model at the site level. However, when a model was instead fit with clustering at the state level, inclusion of random slopes significantly improved model fit for both hippocampal volume (likelihood ratio = 9.93, $p = .007$) and internalizing problems (likelihood ratio = 8.22, $p = .016$), suggesting that state-level characteristics are plausible moderators of the association between log income-to-needs ratio and these outcomes.

Finally, it would be helpful to clarify if the interactions are modeled such that they are actually cross-level interactions between site-level variables (thus accounting for clustering) and income at the individual family level, in order to represent the hypotheses accurately.

Response: As described in the Analysis section, all variables in the interactions were grand-mean centered, and random intercepts and random slopes for the association between log income-to-needs ratio and each of the outcomes were included in those models (and nested within site). This analytic approach certainly accounts for clustering. However, it should be noted that, because log income-to-needs ratio was grand-mean centered and *not* centered at the site level, these are technically not purely cross-level interactions, as the within-site and between-site variance in the relation between log income-to-needs ratio and each of the outcomes are not isolated. This was intentional, as from our perspective, the distinction between within-site variability in income (i.e., family income relative to the other adolescents at that site as indicated by the within-site centered log-income-to-needs ratio) and between-site variability in income (i.e., the average log-income-to-needs ratio of families at that site) is not theoretically meaningful in the context of these analyses.

Finally, please run sensitivity analyses without sites from CA and FL, to determine if results are robust to omitting cases where there are multiple sites per state (give the issue, that should be discussed, in which state-level data is applied to sites though they may differ in social and economic characteristics from the state as a whole).

Response: Respectfully, we disagree with the logic that excluding sites from California and Florida would be informative in determining whether state-level data can be accurately applied to specific locations within the state. To the contrary, it seems that including more participants from more parts of the state would bolster the validity of a state-level variable in being able to generalize to that broader, more diverse sample, as compared to a single site that may or may not be representative of the state as a whole. From our view, excluding over 3,000 participants in 6 sites in 2 states would largely be a test of the robustness of our analyses to a massive reduction in

degrees of freedom for the current analyses. However, in response to this suggestion, we did rerun our analyses including only the site with the most participants in each state (and therefore excluding 1 site in Florida and 3 sites in California). The results of these analyses were consistent with our original findings. The 3-way interactions between cost of living, cash benefits, and income in relation to hippocampal volume ($B=-3.12$, $SE=.826$, $p<.001$), between cost of living, Medicaid expansion, and income in relation to hippocampal volume ($B=-.959$, $SE=.261$, $p<.001$), and between cost of living, cash benefits, and income in relation to internalizing problems ($B=.0365$, $SE=.0171$, $p=.033$) were all significant, and the 3-way interaction between cost of living, Medicaid expansion, and income in relation to internalizing problems was marginal ($B=11.3$, $SE=6.01$, $p=.060$). These results do not suggest that the states with multiple sites had an undue influence on the results.

Some additional questions arise from the moderation analyses, which as the authors note is could be limited the state-level sample size (17). In particular, when considering simply the intersection between cost-of-living and generosity of support, it appears (especially from Figure 2) that there is very minimal representation of (a) low cost of living but generous programs (0 or 1 state, and (b) for Medicaid in particular, high cost of living but expanded support (this is also true for cash assistance at the higher end for cost of living). This raises concerns about the stability of this interaction, and whether it would be stronger instead for focus primarily on two-way interactions, either for each variable separately, or for a combined variable that represents the primary categories. In addition, it wasn't clear whether the two-way interaction model was tested first, and then the three-way interaction. Given the potential challenge with the distribution of categories, it would help to build this model sequentially and determine whether adding the three-way interaction actually improves the model fit over the two-way interaction models. For all such moderation analyses, it is also important to report on all the main effects and interactions in the model so that the overall pattern can be interpreted, rather than only the significant parameters.

Response: The correlation between cost of living and the generosity of antipoverty programs poses an additional challenge to effectively visualizing and decomposing the 3-way interactions between cost of living, antipoverty programs, and log income-to-needs ratio in a way that does not include estimates that fall outside the actual range of values in the data. However, as we demonstrate more clearly in this draft, the models with the 3-way interactions clearly describe the data most accurately. We have revised the figures to better reflect variability in the data that allows us to decompose the influences of cost of living, antipoverty programs, and their interactions (See reproductions of Figures 2B and 3B below).

We have run the models with just the 2-way interactions as the Reviewer suggested and have not found any significant 2-way interactions between cost of living and family income or antipoverty programs and family income in relation to any of the outcomes. We now report this in the text of the manuscript and provide full model outputs in the supplemental materials. Inclusion of the three-way interactions significantly improves the fit of all the models (All Likelihood ratios > 8.90, all $p < .012$) but one. The model with the 3-way interaction between cost of living, cash benefits, and income in relation to internalizing problems marginally improves model fit over the model with just the 2-way interactions between cost of living and income and between cash benefits and income (Likelihood ratio = 5.38, $p = 0.068$). We consider this combined with the statistical significance of the 3-way interactions to be sufficient justification for inclusion of those interaction terms. Finally, as suggested by the Reviewer, we have included regression tables in this draft for all the parameters of interest in the models tested.

In addition, with respect to analyses, it is important to account for family-level nesting within sites in ABCD, which was not mentioned. This could be done either through random exclusion of family members or accounting for family-level clustering in the mixed modeling approach. Additionally, it is important to control for race/ethnicity, particularly given that this variable is included at the site level.

Response: We thank the reviewer for raising this important point. While including random effects for family proved untenable, as the models all failed to converge, we reran analyses with random exclusion of family members, and results remained unchanged. We also control for race/ethnicity at the site and individual levels and find our results are not meaningfully changed. We include these results in supplemental materials.

Moreover, it was surprising not to see parental education utilized as a covariate as well, or an affirmative and strong argument made for its exclusion.

Response: The goals of our analyses were not to determine the specificity of the association of family income with any of our outcomes. Given that parental education and family income are strongly correlated (including in this sample, $r = .61$), including parental education as a covariate in analyses would fundamentally alter the meaning and interpretation of the association between family income and any outcomes to something that is less interpretable or meaningful for our questions. We now state that explicitly in the methods.

Additional

- The timing of the data used in the longitudinal study was not clear. It says “We drew data from the Year 1 assessment”, but then much data actually appears to be from baseline though this is not always specified. The authors should separate the data release version from the year of assessment, clarify which measures come from which time points in the study and why, and thus which analyses are cross-sectional and which are longitudinal.

Response: All data are now from the baseline session, and all data are cross-sectional. We initially included stressful life events as an outcome, which was collected at the year 1 follow-up assessment, but we have removed those analyses from the current manuscript so that all assessments were drawn from the same time point.

- The focus of the title on “brain structure” suggests a wider or more pervasive, global pattern. It would be better to focus more specifically on hippocampal volume, as it is the only aspect of the brain that was investigated.

Response: We agree and have changed the title to “Antipoverty programs mitigate socioeconomic disparities in hippocampal volume and internalizing problems among U.S. youths.”

- Additional papers have been published using ABCD that examine how effects vary by site, and thus the paragraph in the discussion of the methodological innovation should review this literature and place the current paper in context.

Response: We have revised the discussion to acknowledge the literature demonstrating site-level variability in outcomes in the ABCD study. On p. 14: “However, our results, together with other recent studies (Hatzenbuehler et al., 2021), demonstrate that the magnitude of individual-level associations between environmental factors, like family income, and adolescent outcomes, like brain structure and psychopathology, vary systematically as a function of characteristics of the macro-social context.”

Reviewer #2 (Remarks to the Author):

In this study, ABCD study data was utilised to address the question of whether anti-poverty

policies/cash benefits for lower SES families reduced the association between SES and hippocampal volume/internalizing/externalizing symptoms in children. Results were supportive of hypotheses. This is a very interesting study with important implications. I have a few comments/suggestions for changes:

1. Abstract: Please specify that the sample is from the US. Throughout: it should be made clear that much of the discussion is, or may be, only relevant to the US population. Some US-specific things are discussed but not explained, e.g., Medicaid

Response: We have now made it more clear throughout that the sample is from the U.S. and added additional explanations of U.S. policies.

2. Introduction: Some statements are overly speculative, e.g., “Increasing evidence demonstrates that SES can influence the developing brain” and “Childhood SES has consistently been found to influence brain structure”. Does the evidence point to SES having a causal influence on the brain? My understanding is that the existing studies descriptive/correlational.

Response: We have removed causal language throughout the paper.

3. “This finding suggests that reductions in stress exposure may be one pathway through which state-level anti-poverty programs exert a buffering effect on hippocampal volume and internalizing problems in high cost of living states.” This may be true, but the authors have not tested the relevant (mediation) model to address this question. I understand that mediation is not ideal for cross-sectional data, but if there is a theory there, then I think a moderated mediation model would be useful to provide support for the authors’ claim (with appropriate acknowledgement of the limitations of cross-sectional data).

Response: We appreciate this suggestion. In future work, we agree it will be valuable to take advantage of the longitudinal data in the ABCD study to evaluate associations between family income, stress, brain structure, and mental health, as well as sources of regional variability in those associations. For the original analyses, our potential mediator between family income and both internalizing problems and hippocampal volume was stressful life events, which was first assessed at the year-1-follow-up assessment, meaning the mediator in the analyses was actually assessed 1 year *after* the outcomes and may very well reflect events that occurred after the outcomes were assessed. We therefore would consider mediation with those time points inappropriate and difficult to interpret with the causal structure they would imply. The ABCD public data releases currently include data from the 2-year follow-up assessment, at which both psychopathology and hippocampal volume were assessed. However, as detailed in our next response, the collection of these data were heavily impacted by the COVID-19 pandemic, and resulted in attrition that was high and differed as a function of the primary variables of interest in our analysis. We anticipate that will not be the case for future waves of data, which will allow for more valid mediation analyses to be performed, and we fully intend to pursue those analyses when the data are available. Therefore, given the potential for more rigorous investigation in future work, we have decided to remove stressful life events from the current analyses and focus instead on highlighting and demonstrating sources of regional variation in the concurrent associations between family income and both hippocampal volume and internalizing problems.

4. What is the rationale for using data from the year 1 assessment only? Is this the 1-year follow-up?

Response: Only baseline data is used in the current analyses. Family income and hippocampal volume tend to be highly stable; indeed, in ABCD, the correlation of hippocampal volume at baseline with hippocampal volume at the year 2 follow-up is $r=.92$, and the correlations between

log income-to needs-ratio at baseline with the year 2 follow-up is $r=.86$. Further, much of the two-year follow-up assessment, in which the second wave of neuroimaging data was collected, occurred during 2020 and was highly impacted by the COVID-19 pandemic. Participants missing data at the year 2 follow-up had significantly lower income at the baseline assessment (income-to-needs ratio of 3.67 for participants who participated in the year 2 follow up vs. 3.17 for those who did not, $p<.001$) and significantly smaller hippocampal volume at the baseline assessment (8142 mm^3 for participants who completed the year 2 assessment vs. 8083 mm^3 for those who did not, $p=.006$). This type of differential attrition produces data that are missing not at random, which poses a significant challenge for the types of longitudinal analyses we would be conducting. The contribution of demonstrating the potential sources of regional variability in the associations of family income with hippocampal volume and mental health is considerable, so we have chosen to focus this paper on that question. We hope and expect that in years to come, the future waves of ABCD data will make it possible to examine the longitudinal associations between family income, stressful life events, and hippocampal volume, over time and how these longitudinal associations vary between sites with different macrostructural contexts, with fewer challenges related to differential missing data during the pandemic as data collection continues in this sample. However, for the current analyses, there is not a plausible argument that the moderators in our analyses can be caused by our outcomes, and the stability of our independent (income) and dependent (hippocampal volume) variables is such that the current longitudinal data available may not be particularly useful in disentangling the direction of their causal associations.

Reviewer #3 (Remarks to the Author):

Review of Weissman et al.: Antipoverty programs mitigate socioeconomic disparities in brain structure and psychopathology among U.S. youths

In this report, the authors set out to assess whether public policies that increase financial resources for families with low income can reduce socioeconomic disparities in brain development and mental health using the ABCD cohort which spans 21 sites across 17 states in the US. They operationalized antipoverty programs at the state level, apparently essentially assuming that every family within each of the 17 states were availed of monthly Earned Income Tax Credit (EITC; \$266-\$399/month) and/or Temporary Assistance for Needy families (TANF \$286-\$789/month), with some states being more generous in financial assistance than others. They hypothesized lower SES to be associated with smaller hippocampal volume, higher internalizing and externalizing problems, and greater exposure to stressful life events, and that antipoverty programs would mitigate and decrease these associations, particularly in high cost of living states. Based on statistical analyses to test these hypotheses, the authors conclude that the magnitude of previously observed associations between brain and SES did vary as a function of state-level macro-economic factors—including the cost of living and generosity of anti-poverty programs, consistent with their hypotheses. While findings of this nature could have significant implications for public policy in improving the lives of disadvantaged children, there are numerous issues with the analyses and interpretation of results that indicate further investigation is required to draw such conclusions.

Specific comments:

1. The primary concern is that when using metrics at the state level, assumptions are made about similarity in demographics among the various 21 ABCD cohorts that are not supportable. While the ABCD cohort as a whole largely reflects the demographics of the US population, at the site level, they do not. For example, there are 4 distinct sites in California, 2 in Los Angeles (USC/CHLA and UCLA), 1 in the Stanford area of northern California, and one in San Diego. The CHLA site has proportionally

the largest number of very low-income families, while, the Stanford site has proportionally the largest number of very high income families in the ABCD cohort. Both Los Angeles and Stanford have a very high cost of living, equating the populations within the ABCD cohort at those 2 sites would be misleading. While the cost of living is undeniably lower in cities like Tulsa, Oklahoma and Salt Lake City, Utah than the California sites, the proportion of underrepresented minorities is also much lower at those 2 sites. While it is true that some metrics of population characteristics at the state-level could be meaningful and informative, additional family income from antipoverty programs considered as equal in Stanford and Los Angeles are probably not.

Response: These are reasonable critiques that we now acknowledge with more elaboration in the discussion (e.g., on p. 15). The Federal Bureau of Economic Analysis does compute cost of living at the level of the Metropolitan Statistical Area as well as the state level, and we considered using this measure instead of the state-level measure. We ultimately chose to use the state-level measure for several reasons. First, each site was a regional center recruiting between 300 and 1000 participants from a wide catchment area that in most cases spanned well beyond the metropolitan statistical area in which the site was located. Second, in general, the costs of living in the metropolitan areas where the data was collected track closely with the costs of living in the states where the data were collected ($r=.73$ across the 21 sites). Perhaps most importantly, the anti-poverty benefits that are the focus of the paper are policies enacted at the state level, not metropolitan area. Using cost of living estimates from the state level ensures that the fixed effects in the model are drawn from the same geographical level of analysis. Taken together, we did not think using metropolitan area cost of living would be a more accurate representation of the macrostructural conditions in which the families in the ABCD study lived than using state-level cost of living, and there is an advantage for the interpretation and visualization of the results in using the state-level estimates.

We acknowledge that the example the reviewer provides of the California samples is important, but we also want to point out that California is already at the extreme high end of the states included in the analysis in both cost of living and the generosity of its antipoverty programs. The cost of living estimates for the Los Angeles metro area are about the same as the estimates for the state as a whole, while the cost of living for the San Jose area is higher. While using the higher estimates for the San Jose area might be more accurate for many families at that site, given where California is already situated in the distribution, this adjustment may give the California sites an inappropriately large influence in the regression analyses. The more conservative state-level estimates are therefore more appropriate from our perspective. In response to another reviewer comment, we also reran our analyses including only the site with the most participants in each state (and therefore excluding 1 site in Florida and 3 sites in California). The results of these analyses were consistent with our findings. The 3-way interactions between cost of living, cash benefits, and income in relation to hippocampal volume ($B=-3.12$, $SE=.826$, $p<.001$), between cost of living, Medicaid expansion, and income in relation to hippocampal volume ($B=-.959$, $SE=.261$, $p<.001$), and between cost of living, cash benefits, and income in relation to internalizing problems ($B=.0365$, $SE=.0171$, $p=.033$) were all significant, and the 3-way interaction between cost of living, Medicaid expansion, and income in relation to internalizing problems was marginal ($B=11.3$, $SE=6.01$, $p=.060$). These results do not suggest that that differences in cost of living across sites in states with multiple sites had a meaningful influence on the results. Finally, we do control for site-level demographics (race and ethnicity) in our analyses, and we now include sensitivity analyses controlling for race and ethnicity at the individual level in supplemental materials, and find that their inclusion does not influence the pattern of results.

2. The conceptual significance of the results are conveyed as the percent difference in the relationship between brain volume and income to needs with and without accounting for antipoverty benefits (I

think?). For example, the authors report that “among adolescents living in states with higher cost of living, the association between family income and hippocampal volume was about 37% smaller in states with more generous cash assistance programs compared to states with less generous cash assistance programs.” The problems with this statement include that it seems somehow like they are saying that the hippocampus is 37% smaller, but they are talking about the statistical correlation (I think), which leaves me wondering 37% of what?

Response: We appreciate the Reviewer’s thoughtful critique of the way we communicated effect sizes. Our intention was to communicate the effect size of a 3-way interaction in a manner that was as meaningful as possible, but we understand how this came across as misleading. In the latest revision, we have conducted simple slopes analyses that allow us to visualize and explain the unstandardized effect sizes in the native units of the outcome measures instead of in terms of percentages of an unstandardized coefficient, which as the Reviewer points out is not particularly meaningful. Please see the updated description below, which we think more accurately reflects the magnitude of the reported associations.

“Low-income participants living in states with high cost of living and high cash benefits have hippocampal volumes that are, on average, 59 mm³ larger and internalizing problem t-scores that are 1.13 points lower than low-income participants living in states with high cost of living and low cash benefits. On average, the difference in hippocampal volume between low- and high-income participants in high cost of living states with low cash benefits is 212 mm³, but only 141 mm³ in states where cost of living and cash benefits are both high. Thus, more generous cash benefits are associated with income disparities in hippocampal volume that are about 30% lower in states with high cost of living.”

3. In reading the report, I keep thinking that adding government assistance from antipoverty measures could be just like additional family income. With that higher income, of course the oft-observed relationship between family income and brain volume would be attenuated because the income is essentially higher with higher antipoverty cash benefits. Further, equating additional income for all participants in each state within the ABCD cohort doesn’t make sense either because we know from previous reports within the whole ABCD cohort that additional increased increments of income make a much bigger impact on SES brain volume relationships in the lower income than higher income families. So, for example, an additional \$600/month would make the difference between paying rent or buying food for a family making \$50k/year but might pay for a summer vacation for a family making \$150k/year (see Gonzalez et al., 2020, *Frontiers in Human Neuroscience* Figure 1 for relationships between income to needs and total brain volume and total cognition scores in the entire ABCD baseline cohort).

Response: As the Reviewer suggests, we agree that cost of living and antipoverty programs should only influence our outcome variables for people in the low-income range who are actually eligible for benefits (please also see our replies to a similar concern raised by Reviewer 1). It is possible that by using a continuous measure of family income as our independent variable, our effects might be driven by participants who could not benefit from antipoverty programs. However, we can provide evidence that it is indeed the participants at lower income levels who are specifically impacted by these macrostructural differences. We demonstrate this by conducting a region of significance analysis breaking down the 3-way interaction between log income-to-needs ratio, cost of living, and antipoverty programs. These analyses demonstrate that the interactions between cost of living and antipoverty programs are only significant for families with low income. This is displayed in Figure 2

of the revised manuscript and reproduced below. We now report this region of significance analysis in the revised manuscript.

4. There are also site level differences in the number of family members living in the home which could make income-to-needs a bit skewed, I think. The authors do not mention anything in the methods or supplementary figures about how they are dealing with multiple siblings or even the twin cohorts in their analyses. Don't families with multiple children get government assistance for each child? So, if a family has 2 children, they would get twice the monetary assistance of a family with only one child?

Response: The site-level differences in the number of family members living in the home are small. The median family size is 4 in the sample as a whole and in every site but Utah, where the median family size is 6. The mean family size is 4.68 in the sample as a whole and ranges from 4.23 in Los Angeles to 5.68 in Utah (the next highest is 5.02 in Virginia). There is no association of cost of living or the generosity of antipoverty programs with family size. With regard to the role of family size in receipt of benefits, family size is factored into eligibility for benefits as well as the benefit amount based on calculations that are state specific. However, in no state are benefits allocated uniformly on a per-child basis, which was the case for the short-lived Child Tax Credit. Family size is also, of course, factored into the calculation of income-to-needs ratio. Thus, while family size factors into both the operationalization of family income as well as the size of the benefit received by a given family, there is no evidence that site-level variability in family size is likely to be a confounder in these analyses. With regard to families that had multiple siblings (twins) in the ABCD sample, including random effects for family proved untenable, as the models all failed to converge. However, we reran analyses with random exclusion of siblings, and results remained unchanged. We now include these results in supplemental materials.

5. It is not at all clear why the authors account for race differences at each site as the proportion of the sample at that study site that was White, Black, and Latinx, rather than using that variable at the individual level of participant self-report of race/ethnicity.

Response: Because we are looking at site-level moderation of the association between family income and each of our outcomes, we thought that the site-level demographics would be plausible confounders. Conversely, it is our view that using individual-level racial categories as variables of interest or covariates presumes a biological basis for these racial categories that is not supported by evidence (see Helms et al., 2005 for extensive discussion of this issue). However, in the interest

of demonstrating that our results hold whether race and ethnicity are included as site-level or individual-level variables, we have conducted additional sensitivity analyses controlling for race and ethnicity at the individual level, and our results remain virtually unchanged. We have added these sensitivity analyses to supplemental materials.

Minor concerns:

6. On multiple occasions in the manuscript, the authors use the term Adolescent Behavior and Cognitive Development (ABCD) study.

Response: Thank you for flagging this! We have fixed this mistake.

7. Using just one region of interest, hippocampal volume (statistically corrected for ICV) does not show any specificity to that particular region.

Response: This is true, and accordingly we do not argue for specificity. Indeed, the associations between SES and brain structure tend to be widespread throughout the entire brain (e.g., Noble et al., 2015). We chose the hippocampus because the association between SES and hippocampal volume is among the most well replicated, and the mechanisms that contribute to that association have been identified in a way that is not true for other brain regions (Luby et al., 2013; Luby et al., 2012; Pagliaccio et al., 2014). Therefore, we considered hippocampal volume to be a good starting point for establishing this novel approach to evaluating the impacts of macrostructural characteristics on brain structure.

8. There are many geocoded measures at the census tract level that could be used to estimate cost of living at the individual participants home address rather than having a single cost of living measure collapsed across all participants who live in each state.

Response: Our research team does not have access to the participants' home addresses or census tracts, which would be necessary to do what the Reviewer suggests here, and the geocoded data that is released does not include information about cost of living. Further, even if we were able to use participants' geocoded addresses in this way, doing so would lead to considerable complications to the models. Because cost of living would vary within sites, it would become an individual-level variable, and therefore we would need to include a random slope of cost of living in all the models. This would likely lead to problems with model fit and convergence that in our view outweigh the potential benefits, given that the costs of living within metropolitan areas tends to be highly correlated with the cost of living of the state ($r=.73$ across the 21 sites).

Reviewer #4 (Remarks to the Author):

Summary: This study uses national data from the multisite ABCD study to examine the association between individual-level and state-level socioeconomic factors with brain development and mental health. ABCD is a rich data set, and evaluating the effects of anti-poverty policies is critically important. The manuscript is also well written and the figures are clear and well labeled. However, the fundamental flaw of this manuscript is that the correlational design of the current analysis is overly simplistic, making it not particularly rigorous or well-suited to address the multitudinous sources of confounding. That is, simply regressing each outcome on the exposure of interest while adjusting for various covariates does not provide strong support for the hypothesis that these exposures matter for these outcomes. Here are additional details on my methodological concerns:

1. The individual-level SES variable was simply constructed using just income and household size. The

exposure-outcome relationship could be confounded by any number of unmeasured SES or other confounders, including experiences of racism, wealth, housing, immigration status, etc. There is a rich literature in social epidemiology about the importance of these many operationalizations of SES.

Response: While we agree with the reviewer that SES is complex and multifaceted, our modeling decision was informed by a recent meta-analysis demonstrating that the associations between various indicators of SES (e.g. family income, poverty, parental education) and mental health are distinct, but that they are stronger predictors of mental health when treated independently than when combined into a composite score (e.g. the Hollingshead index) (Peverill et al., 2021). For the purposes of this analysis, we examine family income as our primary variable of interest because this is the measure of SES that is most directly influenced by the macrostructural characteristics we investigate here. That is not to say that it is the best or most important measure of SES. We appreciate the Reviewer raising this point, and we have revised the manuscript to more precisely reference the construct investigated here as well as to acknowledge that family income is just one facet contributing to a much more complex construct.

2. Similarly, the state-level variables are a small selection of policies that are likely to be confounded by many other characteristics of states that are likely to affect both the exposure and outcome. The University of Kentucky Center for Poverty Research has a rich database of other types of state characteristics and policies that are commonly included as covariates in studies like this: <https://ukcpr.org/resources/national-welfare-data>

Response: We conducted extensive sensitivity analyses controlling for 10 other macrostructural characteristics – including population density, economic inequality, tightness/looseness, political preferences, incarceration rate, unemployment rate, enrollment in state funded preschool, women’s political participation, reproductive rights, and 4th grade reading levels among low-income students. We controlled not only for the main effects of these potential confounders, but also their interaction with family income. In all cases, our results held after adjustment for these wide-ranging social, political, economic, and educational factors that could be plausible confounders. We are grateful for the recommendation that we consult the resource provided by the University of Kentucky Center for Poverty Research to identify other relevant state-level covariates. In reviewing this resource, we found that our sensitivity analyses had already controlled for, or included as a moderator, all the characteristics included in this database that varied at the state level. As such, we have addressed this Reviewer concern through extensive sensitivity analysis and demonstrate that our results are robust to control for wide-ranging state-level characteristics, including all potential confounders suggested by the Center for Poverty Research resource suggested. We would be happy to conduct additional sensitivity analyses if the Reviewer identifies other specific confounders they would like us to test.

3. Furthermore, the state variables are ecological in nature, i.e., state-level averages of typical benefit levels. There is no attempt to restrict the sample to those individuals for whom cash assistance or safety net policies matter most, e.g., racial/ethnic marginalized communities, low-income individuals, etc. Indeed, it would be worthwhile to test whether there is an association between these exposures/outcomes among high-income groups, which, if found, would support the idea that the relationship is strongly confounded.

Response: As the Reviewer suggests, we agree that cost of living and antipoverty programs should only influence our outcome variables for people in the low-income range who are actually eligible for benefits (please also see related replies to Reviewers 1 and 2). It is possible that by using a continuous measure of family income as our independent variable, our effects might be driven by participants who could not benefit from antipoverty programs. However, we can provide evidence that it is indeed the participants at lower income levels who are specifically impacted by these macrostructural differences. We demonstrate this by conducting a region of significance analysis breaking down the 3-way interaction between log income-to-needs ratio, cost of living, and antipoverty programs. These analyses demonstrate that the interactions between cost of living and antipoverty programs are only significant for families with low income. This is displayed in Figure 2 of the revised manuscript and reproduced below. We now report this region of significance analysis in the revised manuscript.

4. On that note, there is no table of sample characteristics to indicate the degree of socioeconomic vulnerability in this sample. What percent of the sample is low-income, what is the racial/ethnic make-up, how old are the children, what is the educational level of their parents, etc., etc.?

Response: We provide a table summarizing the income distribution across the states in the sample in supplemental Table S1. Table 1 in the main text summarizes the racial and ethnic demographics of the sample at the state level. All participants in the baseline assessment of the ABCD study are about 10 years old. This information is now provided in the methods (p. 16): “We drew data from the baseline assessment (ABCD 3.0) of 11,864 youth with data on parent-reported psychopathology and 11,533 youth with brain structure data (mean age 9.91 years, $SD=.622$).”

5. Even if the issues above were addressed, fundamentally this would still be a correlational analysis that does not contribute much new information to the existing literature on poverty and health. This analysis would be strengthened by more rigorously evaluating a single policy using a quasi-experimental technique, as has been done in prior research on the EITC, WIC, etc. E.g., Strully et al, ASR 2010; Hoynes et al JPE 2011.

Response: While we acknowledge some shortcomings of the ABCD data when it comes to evaluating the impacts of public policy, we also want to highlight the uniqueness of the opportunity it provides, and consequentially the great value of these analyses and findings. The ABCD study is the *only dataset* to collect neuroimaging data from large numbers of children and adolescents, including those at the low end of the income distribution, who live in a range of macrosocial contexts that differ with respect to their anti-poverty policies. Indeed, most neuroimaging studies are conducted in a single location, in which everyone is exposed to the same macrosocial environment, precluding the opportunity to link contextual variation in macrosocial factors such as anti-poverty policies to

neural outcomes. While natural experiments, as suggested by the Reviewer, may one day be feasible with this dataset as policy changes at the state level unfold in the future, *this type of design is not possible at this time with the ABCD data because there were no substantial changes in these poverty-relevant policies in the ABCD states during the data collection period.* Nor to our knowledge is it possible to use any other dataset to conduct the type of natural experiment the Reviewer is suggesting as there are no other datasets that have collected harmonized neuroimaging data on children living in different contexts over time to allow policy changes to be examined as predictors of changes in neural outcomes. While it may be possible to conduct the type of study the Reviewer is suggesting at a future date, it is not possible presently. The analysis we present is the first and only attempt to link contextual variation in state anti-poverty policies to brain development in children. As the ABCD study continues, we hope that these types of questions will continue to be examined in different ways to determine whether our results can be replicated in other types of study designs when they become possible to implement. Additionally, we have highlighted the correlational nature of the study as a limitation in the Discussion section of the paper.

6. Alternately, this study could take advantage of the fact that the ABCD study includes multiple waves. The current study only used Wave 1. Using multiple waves and employing an individual-level fixed effects design would strengthen causal inference about the effects of individual-level factors, or a state-level fixed effects design could provide stronger evidence about the effects of time-varying state-level factors. E.g., see Brüderl and Ludwig. "Fixed-effects panel regression." The Sage handbook of regression analysis and causal inference (2015).

Response: Only baseline data are used in the current analyses. This approach acknowledges that family income and policy differences over the 10 years of participants' lives preceding this baseline assessment likely contribute to the individual differences described. Family income ($r=.86$ from baseline to 2-year follow up) and hippocampal volume ($r=.92$ from baseline to 2-year follow up) are highly stable, with changes occurring infrequently or slowly over many years. This poses a significant challenge for examining the effects of policy change over the time course of the ABCD study at the present moment. Further, as we note above much of the two-year follow-up assessment, in which the second wave of neuroimaging data was collected, occurred during 2020 and was highly impacted by the COVID-19 pandemic. Participants missing data at the year 2 follow-up had significantly lower income at the baseline assessment (income-to-needs ratio of 3.67 for participants who participated in the year 2 follow up vs. 3.17 for those who did not, $p<.001$) and significantly smaller hippocampal volume at the baseline assessment (8142 mm^3 for participants who completed the year 2 assessment vs. 8083 mm^3 for those who did not, $p=.006$). This type of differential attrition produces data that are missing not at random, which poses a significant challenge for the types of longitudinal analyses we would be conducting. The contribution of demonstrating the potential sources of regional variability in the concurrent associations of family income with hippocampal volume and mental health is considerable, so we have chosen to focus this paper on that. We hope and expect that in years to come, the future waves of ABCD data will make it possible to examine the longitudinal associations between these factors over time with more timepoints, allowing for greater opportunity for income, brain structure, and public policy to change with fewer challenges related to the causes of differential missing data during the pandemic as data collection continues in this sample.

7. It's unlikely that something like Medicaid expansion could so quickly act to change hippocampal volume, i.e., there is not a plausible mechanism for such a rapid effect. In fact, this finding again suggests that some other source of confounding at the state level is present.

Response: Given that in most states Medicaid was expanded in 2014, 3 years prior to the study, we think expanded Medicaid access is a plausible means through which the impacts of poverty are mitigated in high-cost-of-living states. At the same time, we acknowledge that the macrostructural characteristics examined are intertwined with demographic, historical, economic, and political characteristics that shape regional differences in the structural environments that low-income families navigate. In this revision, we have been careful to communicate more clearly that this paper does not serve as a direct evaluation of specific policies, which would require a natural experiment or randomized controlled trial. Instead, the variables selected in this study serve as quantitative indicators of the broader structural environment in which families with different levels of income are embedded, which we are able to operationalize on a meaningful scale (dollars), and at a level of regional variability (U.S. states) that is both meaningful in the real world (i.e., it is the geographic unit where these policies are implemented) and aligned with the sampling procedures of the ABCD study. For example on p. 5:

“Importantly, other public policies may also mitigate the impact of low SES on child development (Bitler et al., 2017), but their generosity does not vary between U.S. states. These include the Supplemental Nutrition Assistance Program (i.e., food stamps) and, recently, the child tax credit. In addition, policies at the county, city, or school district level may also contribute to an environment that lessens socioeconomic disparities. Because our primary comparisons are across states, we specifically examine the mitigating influence of the largest antipoverty programs that vary by state: EITF, TANF, and Medicaid. Data from natural experiments indicate that these large programs have documented efficacy at improving family functioning, physical health, academic achievement, and overall wellbeing, and at reducing psychological distress.(Averett & Wang, 2018; Baltagi & Yen, 2016; McMorroo et al., 2017; Wang, 2015) However, the correlational analyses described in this study are not intended to serve as evaluations of specific policies. but rather as quantitative indicators of the broader structural environment in which families with different levels of income are embedded, which we are able to operationalize on a meaningful scale (e.g. dollars) and at a meaningful level of regional variability (U.S. states).”

References

- Averett, S., & Wang, Y. (2018). Effects of Higher EITC Payments on Children's Health, Quality of Home Environment, and Noncognitive Skills. *Public Finance Review*, *46*(4), 519–557.
<https://doi.org/10.1177/1091142116654965>
- Baltagi, B. H., & Yen, Y.-F. (2016). Welfare Reform and Children's Health. *Health Economics*, *25*(3), 277–291. <https://doi.org/10.1002/hec.3139>
- Bitler, M., Hoynes, H., & Kuka, E. (2017). Child Poverty, the Great Recession, and the Social Safety Net in the United States. *Journal of Policy Analysis and Management*, *36*(2), 358–389.
<https://doi.org/10.1002/pam.21963>
- Hatzenbuehler, M. L., Weissman, D. G., McKetta, S., Lattanner, M. R., Ford, J. V., Barch, D. M., & McLaughlin, K. A. (2021). Smaller Hippocampal Volume Among Black and Latinx Youth Living in High-Stigma Contexts. *Journal of the American Academy of Child & Adolescent Psychiatry*, *0*(0). <https://doi.org/10.1016/j.jaac.2021.08.017>
- Helms, J. E., Jernigan, M., & Mascher, J. (2005). The meaning of race in psychology and how to change it: A methodological perspective. *The American Psychologist*, *60*(1), 27–36.
<https://doi.org/10.1037/0003-066X.60.1.27>
- LaHuis, D. M., & Ferguson, M. W. (2009). The Accuracy of Significance Tests for Slope Variance Components in Multilevel Random Coefficient Models. *Organizational Research Methods*, *12*(3), 418–435. <https://doi.org/10.1177/1094428107308984>
- Luby, J., Belden, A., Botteron, K., Marrus, N., Harms, M. P., Babb, C., Nishino, T., & Barch, D. (2013). The Effects of Poverty on Childhood Brain Development: The Mediating Effect of Caregiving and Stressful Life Events. *JAMA Pediatrics*, *167*(12), 1135–1142.
<https://doi.org/10.1001/jamapediatrics.2013.3139>
- Luby, J. L., Barch, D. M., Belden, A., Gaffrey, M. S., Tillman, R., Babb, C., Nishino, T., Suzuki, H., & Botteron, K. N. (2012). Maternal support in early childhood predicts larger hippocampal

volumes at school age. *Proceedings of the National Academy of Sciences*, 109(8), 2854–2859.

<https://doi.org/10.1073/pnas.1118003109>

McMorrow, S., Gates, J. A., Long, S. K., & Kenney, G. M. (2017). Medicaid Expansion Increased Coverage, Improved Affordability, And Reduced Psychological Distress For Low-Income Parents. *Health Affairs*, 36(5), 808–818. <https://doi.org/10.1377/hlthaff.2016.1650>

Pagliaccio, D., Luby, J. L., Bogdan, R., Agrawal, A., Gaffrey, M. S., Belden, A. C., Botteron, K. N., Harms, M. P., & Barch, D. M. (2014). Stress-System Genes and Life Stress Predict Cortisol Levels and Amygdala and Hippocampal Volumes in Children. *Neuropsychopharmacology*, 39(5), 1245–1253. <https://doi.org/10.1038/npp.2013.327>

Peeverill, M., Dirks, M. A., Narvaja, T., Herts, K. L., Comer, J. S., & McLaughlin, K. A. (2021). Socioeconomic status and child psychopathology in the United States: A meta-analysis of population-based studies. *Clinical Psychology Review*, 83, 101933.

<https://doi.org/10.1016/j.cpr.2020.101933>

Shrivastava, A., & Azito Thompson, G. (2022). *Policy Brief: Cash Assistance Should Reach Millions More Families to Lessen Hardship* [Policy Brief]. Center on Budget and Policy Priorities.

<https://www.cbpp.org/research/family-income-support/cash-assistance-should-reach-millions-more-families-to-lessen>

Wang, J. S.-H. (2015). TANF coverage, state TANF requirement stringencies, and child well-being. *Children and Youth Services Review*, 53, 121–129.

<https://doi.org/10.1016/j.childyouth.2015.03.028>

REVIEWER COMMENTS

Reviewer #1 (Remarks to the Author):

Overall the revised papers has been considerable strengthened, though some issues remain.

The authors response does an excellent job both clarifying and acknowledging that they are not able to evaluate specific policies or establish that anti-poverty policies themselves are the buffer. They acknowledge this by noting: ""In this revision, we have been careful to communicate more clearly that this paper does not serve as a direct evaluation of specific policies, which would require a natural experiment or randomized controlled trial. Instead, the variables selected in this study serve as quantitative indicators of the broader structural environment in which families with different levels of income are embedded". However, the paper is still written in many sections as if the antipoverty programs themselves – rather than the broader structural environment – are the moderators. This is most obviously salient in the title itself, which states "Antipoverty programs mitigate socioeconomic disparities...". Multiple sentences in the abstract (first and last, for example), the last sentence of the first introductory paragraph, and many other sections still focus specifically on anti-poverty policies themselves, not on the broader structural environment. A detailed revision is needed throughout the paper to focus on antipoverty programs at the state level as an index of the broader structural environment.

The authors have done well to clearly indicate, via both simple slopes and sensitivity analyses, that they moderation effects are limited to low-income families who could actually benefit from these policies – this is very important and a key contribution of the paper. To strengthen this, though, it is important to include both the simple slopes analysis but also to report the sensitivity analyses of dichotomous indices using poverty as well as 5x the poverty lines are very helpful and, at a minimum, which should be included in supplementary material at a minimum. Given that the authors argue for the importance of operationalizing the structural environment in a manner meaningful in the real world, the same argument can be made for a focus on a dichotomous indicator that specifically captures how such moderators would function in the real world.

The argument that the income bins in ABCD are too broad for antipoverty program to make an appreciable impact is not convincing, particularly in terms of the lower-income categories. Consequently, it is important to discuss the fact that the measure already incorporates estimates of benefit receipt needs as a limitation in the discussion. This concern, though, will be somewhat mitigated by the inclusion of sensitivity analyses with a dichotomous indicator – while this indicator still is sensitive to benefit receipt that moves families out of poverty, it is not sensitive to benefit receipt that increases income but below the poverty line, highlighting the importance of including these analyses

Please include, in the supplement, the sensitivity analyses utilizing only one site from Florida and California, and describe in the text as done in the response letter.

Please separate the ABCD release version from the discussion of assessments – these are different (ABCD 3.0 does not necessarily clarify that all data are baseline data).

Reviewer #2 (Remarks to the Author):

The authors have addressed all of my concerns.

Reviewer #3 (Remarks to the Author):

I still think the overall impression in reading the manuscript is that it does not go much beyond previous reports of brain, cognition family income (INR) studies published with ABCD data and other large data sets which show strong associations at the individual level (as opposed to the state level). Notably, these associations are non-linear, strongest among families with lower income to needs, and, pretty much flat as INRs go up beyond ~400% of the federal poverty level (see Gonzalez et al <https://pubmed.ncbi.nlm.nih.gov/33192411/> for analyses of ABCD baseline data, and Noble et al <https://www.nature.com/articles/nn.3983> for analyses in an independent sample from the PING study). Essentially, the findings from the state level described here replicate these non-linear effects with essentially a sample size of 17 (given all the metrics that assume all participants in California, for example, are the same on demographics, and, that they are different from all participants in Utah, for example). It is not clear to me that the anti-poverty programs are making the difference in brain and internalizing problems.

At the very least, the title and abstract need to reflect that they are making these assumptions based on state-level average metrics, and, not individual level actual benefits.

Reviewer #4 (Remarks to the Author):

The authors have been responsive to reviewer comments.

We appreciate the thoughtfulness with which you and the Reviewers considered both the merits and outstanding questions of the analyses and interpretations implemented in this study. Responses to individual comments appear in **bold** below.

First, to both reviewers, please note that in checking all of our code for inputting and analyzing the data to ensure that all results were reproduced exactly, we identified an error that led 11 participants to be included in analyses related to hippocampal volume who needed to be excluded. The T1w structural MRI data for these participants are listed as needing to be excluded for data quality based on the ABCD criteria as described in the methods and indicated in the data release. https://nda.nih.gov/data_structure.html?short_name=abcd_imgincl01, however in reproducing all the steps of the dataset construction process, we discovered that for an unknown reason they were not excluded as they should have been. Correcting this error altered the values in the results minimally but did not change the findings or conclusions. The main effect of log-income-to-needs ratio on hippocampal volume is now reported on p. 8: “ $(t(9,888)=7.78, p<.001, B=62.71, 95\% CI=46.91 \text{ to } 78.52, \beta=.079$. When those participants included these statistics were: $(t(9,896)=7.71, p<.001, B=62.61, CI=48.46 \text{ to } 79.86, \beta=.079)$. The tables of results of analyses examining the 3-way interaction between log-income-to-needs ratio, cost of living, and hippocampal volume (Table 2) before and after this correction are reproduced for reference below.

Table 2: Results of Moderation Analyses (Current, excluding 11 additional participants)

Hippocampal Volume							
Cash Benefits				Medicaid Expansion			
	B	SE	p		B	SE	p
Income	82.0	9.66	<.001	Income	95.8	16.0	<.001
Cost of Living (COL)	-134	435	.761	COL	-278	614	.657
Cash Benefit (Cash)	-.032	.217	.885	Medicaid Expansion (ME)	-15.4	40.8	.712
Income x COL	242	174	.164	Income x COL	736	259	.005
Income x Cash	-.0992	.101	.326	Income x ME	-26.8	18.6	.149
COL x Cash	2.63	2.84	.370	COL x ME	350	649	.599
Income x COL x Cash	-3.64	1.16	.002	Income x COL x ME	-983	297	.001

Table 2: Results of Moderation Analyses (Original, including 11 participants in error)

Hippocampal Volume							
Cash Benefits				Medicaid Expansion			
	B	SE	p		B	SE	p
Income	84.4	9.64	<.001	Income	107.3	18.9	<.001
Cost of Living (COL)	-167	338	.630	COL	-240	502	.639
Cash Benefit (Cash)	.024	.243	.922	Medicaid Expansion (ME)	-10.5	48.2	.830
Income x COL	229	138	.098	Income x COL	613	209	.003
Income x Cash	-.142	.111	.202	Income x ME	-35.4	21.1	.092
COL x Cash	1.90	1.87	.327	COL x ME	314	539	.570
Income x COL x Cash	-2.62	.766	<.001	Income x COL x ME	-824	232	<.001

Reviewer #1 (Remarks to the Author):

Overall the revised papers has been considerable strengthened, though some issues remain.

The authors response does an excellent job both clarifying and acknowledging that they are not able to evaluate specific policies or establish that anti-poverty policies themselves are the buffer. They acknowledge this by noting: ““In this revision, we have been careful to communicate more clearly that this paper does not serve as a direct evaluation of specific policies, which would require a natural experiment or randomized controlled trial. Instead, the variables selected in this study serve as quantitative indicators of the broader structural environment in which families with different levels of income are embedded”. However, the paper is still written in many sections as if the antipoverty programs themselves – rather than the broader structural environment – are the moderators. This is most obviously salient in the title itself, which states “Antipoverty programs mitigate socioeconomic disparities...”. Multiple sentences in the abstract (first and last, for example), the last sentence of the first introductory paragraph, and many other sections still focus specifically on anti-poverty policies themselves, not on the broader structural environment. A detailed revision is needed throughout the paper to focus on antipoverty programs at the state level as an index of the broader structural environment.

We appreciate that the reviewer thinks that the paper has been considerably strengthened as a result of the revisions.

We have edited the manuscript, including the title and abstract, to consistently describe the implications of this paper in terms of moderation by the broader macrostructural context (i.e., the generosity of the social safety net for low-income families), instead of as an evaluation of specific policies. We retain language about the specific policy indicators we measured when describing patterns of results from the analyses we performed which relied on these specific indicators of the broader structural environment. But we have updated our broader framing and interpretations to focus on macrostructural characteristics throughout the paper.

The updated title is: Generosity of state-level social safety net mitigates neurodevelopmental and mental health consequences of poverty in US children

We provide several illustrative examples of these changes below:

Abstract:

“Can macrostructural characteristics, such as cost-of-living and state-level antipoverty programs, alter the magnitude of socioeconomic disparities in brain development and mental health?”

“These findings suggest that state-level macrostructural characteristics, including the generosity of anti-poverty policies, may mitigate the neurodevelopmental and mental health consequences of low income.”

Introduction:

p. 3: “Macrostructural characteristics that influence the material resources available to low-income families may alter the strength of the association between low income and health and neurodevelopmental outcomes.”

p. 6: “The current study builds on this recent work to examine whether cost of living and the generosity of a state’s social safety net for families living in poverty moderate the association of family income with hippocampal volume and mental health outcomes using data from the large, multisite ABCD study.”

p. 6-7 “Importantly, the correlational analyses described in this study are not intended to serve as a direct evaluation of these specific policies. Rather, the policies selected in this study are intended to serve as quantitative indicators of the broader macrostructural environment related to the generosity of the state’s social safety net for families living in poverty, which we are able to operationalize on a meaningful scale (e.g., dollars) and at a meaningful level of regional variability (U.S. states) at which these policies are actually implemented.”

Discussion:

p. 14: “Together, these findings suggest that macrostructural factors related to the generosity of the state’s social safety net for families living in poverty are strongly associated with socioeconomic disparities in hippocampal volume and internalizing problems, and that structural policy interventions may be an effective strategy for reducing these disparities, though this interpretation awaits replication with other research designs (e.g., quasi-experiments).”

p. 15: “Given that we observed the same type of 3-way interaction of family income and cost of living with cash assistance and Medicaid expansion, we expect that these policy differences—while helpful in and of themselves—are likely indicative of a broader collection of policies and structural supports for families facing economic hardship that were not directly measured in this study but reflect the generosity of the social safety net for low-income families (e.g., availability of free or reduced cost early childhood education programs).”

The authors have done well to clearly indicate, via both simple slopes and sensitivity analyses, that they moderation effects are limited to low-income families who could actually benefit from these policies – this is very important and a key contribution of the paper. To strengthen this, though, it is important to include both the simple slopes analysis but also to report the sensitivity analyses of dichotomous indices using poverty as well as 5x the poverty lines are very helpful and, at a minimum, which should be included in supplementary material at a minimum. Given that the authors argue for the importance of operationalizing the structural environment in a manner meaningful in the real world, the same argument can be made for a focus on a dichotomous indicator that specifically captures how such moderators would function in the real world.

We are grateful for this suggestion and have added the results of these supplemental analyses as Table S5 and referenced them in the text on p. 13:

“To further interrogate these associations, we conducted supplementary analyses, where we examined income dichotomously to reflect whether family income was above vs. below the federal poverty line. This allowed us to examine whether the patterns of findings were observed only for participants with lower income who could potentially benefit from these programs but not those with higher incomes who could not. Indeed, the pattern of results was consistent with our original findings, in that the interactions of cost of living, generosity of anti-poverty policies, and family income are all significant. However, if we use a higher cutoff that does not correspond with eligibility for benefits (i.e., dichotomizing above vs. below 5x the federal poverty line), these interactions are not significant for any outcome (Supplemental Table S5), providing further support for our interpretation that these associations are only apparent for individuals who are actually eligible for anti-poverty programs.”

The argument that the income bins in ABCD are too broad for antipoverty program to make an appreciable impact is not convincing, particularly in terms of the lower-income categories. Consequently, it is important to discuss the fact that the measure already incorporates estimates of benefit receipt needs as a limitation in the discussion. This concern, though, will be somewhat mitigated by the inclusion of sensitivity analyses with a dichotomous indicator – while this indicator still is sensitive to benefit receipt that moves families out of poverty, it is not sensitive to benefit receipt that increases income but below the poverty line, highlighting the importance of including these analyses

We have added this to the limitations on p. 18 to address the Reviewer’s point: “Another potential issue is that the parent-reported total family income could include cash assistance received by the family. Consequently, the income measure may already include variance related to the putative moderator. However, when repeating analyses using a dichotomous indicator of poverty, the results were consistent with the current findings. Further, more generous antipoverty programs (particularly cash assistance) may not reduce stress by increasing monthly or yearly income in all cases. For example, these benefits may allow families to make decisions that lead to a decrease in wages but that also reduce stress, such as working fewer hours.”

Please include, in the supplement, the sensitivity analyses utilizing only one site from Florida and California, and describe in the text as done in the response letter.

We now include a description of these supplemental analyses in the text on p. 13.

“In addition, to evaluate whether results were driven by states with multiple sites and a greater number of participants (i.e., California and Florida), we reran our analyses including only the site with the most participants in each state (and therefore excluding 1 site in Florida and 3 sites in California). The results of these analyses were consistent with our original findings (See Supplemental Table S6).”

Please separate the ABCD release version from the discussion of assessments – these are different (ABCD 3.0 does not necessarily clarify that all data are baseline data).

We have edited the description on p. 18 to read: “Data come from data release 3.0 of the ABCD study, the largest study of brain development in the U.S. (<https://abcdstudy.org>). We drew data from the baseline assessment of 11,864 youth...”

Reviewer #3 (Remarks to the Author):

I still think the overall impression in reading the manuscript is that it does not go much beyond previous reports of brain, cognition family income (INR) studies published with ABCD data and other large data sets which show strong associations at the individual level (as opposed to the state level). Notably, these associations are non-linear, strongest among families with lower income to needs, and, pretty much flat as INRs go up beyond ~400% of the federal poverty level (see Gonzalez et al <https://pubmed.ncbi.nlm.nih.gov/33192411/> for analyses of ABCD baseline data, and Noble et al <https://www.nature.com/articles/nn.3983> for analyses in an independent sample from the PING study). Essentially, the findings from the state level described here replicate these non-linear effects with essentially a sample size of 17 (given all the metrics that assume all participants in California, for example, are the same on demographics, and, that they are different from all participants in Utah, for example). It is not clear to me that the anti-poverty programs are making the difference in brain and

internalizing problems.

The reviewer is correct that much research has been devoted to understanding the impact of socioeconomic status (SES) on brain structure and mental health, including within the ABCD study. We do demonstrate that income has nonlinear associations with hippocampal volume and mental health, as the reviewer describes. Excellent work, including Gonzalez et al. (2020), which the reviewer cites, has also been done identifying characteristics of the rearing environment that help to explain this association.

However, research examining how the broader macrosocial context might influence the links between SES and neurodevelopmental and mental health outcomes has not previously been conducted. This is in large part because there has never previously been a dataset like ABCD that includes neuroimaging data collected from large numbers of participants spanning diverse contexts that affords the opportunity to examine geographic variation in these associations. Although it may seem self-evident to the reviewer that macrostructural characteristics that influence the material resources available to families would alter these associations, this is the first study in neuroscience, developmental psychology, or clinical science to evaluate this empirically. The contribution of this paper is not in showing the associations of family SES with hippocampal volume and mental health, but in showing that the magnitude of those associations varies meaningfully as a function of the broader macrosocial environment in which children are being raised.

We now make this more explicit in the introduction on p. 7. “We hypothesized that lower family income would be associated with smaller hippocampal volume and higher internalizing and externalizing problems, consistent with prior work^{2,3,7,11–15,20,29}. We additionally hypothesize that the magnitude of those associations would be increased in states with higher cost of living, but would be smaller in states with more generous anti-poverty programs, particularly in high cost of living states. In doing so we provide a novel test of whether the well-established associations of family income with brain structure and mental health vary as a function of the broader macrostructural environment in which children are being raised.”

At the very least, the title and abstract need to reflect that they are making these assumptions based on state-level average metrics, and, not individual level actual benefits.

We have edited the title and abstract to clarify that the moderators are state-level indicators. We have also made numerous changes to the manuscript throughout to note that we are examining state-level indicators of the broader macrostructural context rather than evaluating specific policies.

The updated title is: Generosity of state-level social safety net mitigates neurodevelopmental and mental health consequences of poverty in US children

We provide several illustrative examples of these changes below:

Abstract:

“Can macrostructural characteristics, such as cost-of-living and state-level antipoverty programs, alter the magnitude of socioeconomic disparities in brain development and mental health?”

“These findings suggest that state-level macrostructural characteristics, including the generosity of anti-poverty policies, may mitigate the neurodevelopmental and mental health consequences of low income.”

Introduction:

p. 3: “Macrostructural characteristics that influence the material resources available to low-income families may alter the strength of the association between low income and health and neurodevelopmental outcomes.”

p. 6: “The current study builds on this recent work to examine whether cost-of-living and the generosity of the social safety net for families living in poverty moderate the association of family income with hippocampal volume and mental health outcomes using data from the large, multisite ABCD study.”

p. 6-7 “Importantly, the correlational analyses described in this study are not intended to serve as a direct evaluation of these specific policies. Rather, the policies selected in this study are intended to serve as quantitative indicators of the broader structural environment in which families with different levels of income are embedded, which we are able to operationalize on a meaningful scale (e.g., dollars) and at a meaningful level of regional variability (U.S. states) at which these policies are actually implemented.”

Discussion:

p. 14: “Together, these findings suggest that macrostructural factors related to the generosity of the social safety net for families living in poverty have a powerful influence on socioeconomic disparities in hippocampal volume and internalizing problems, and that structural policy interventions may be an effective strategy for reducing these disparities.”

p. 15: “Given that we observed the same type of 3-way interaction of family income and cost of living with cash assistance and Medicaid expansion, we expect that these policy differences—while helpful in and of themselves—are likely indicative of a broader collection of policies and structural supports for families facing economic hardship that were not directly measured in this study but reflect the generosity of the social safety net for low-income families (e.g., availability of free or reduced cost early childhood education programs).”